# Molecular Biology, Composition and Physiological Functions of Cuticle Lipids in Fleshy Fruits

**DOI:** 10.3390/plants11091133

**Published:** 2022-04-22

**Authors:** Heriberto García-Coronado, Julio César Tafolla-Arellano, Miguel Ángel Hernández-Oñate, Alexel Jesús Burgara-Estrella, Jesús Martín Robles-Parra, Martín Ernesto Tiznado-Hernández

**Affiliations:** 1Coordinación de Tecnología de Alimentos de Origen Vegetal, Centro de Investigación en Alimentación y Desarrollo A.C., Carretera Gustavo Enrique Astiazarán Rosas 46, Hermosillo 83304, Sonora, Mexico; heriberto.garcia.dc19@estudiantes.ciad.mx; 2Laboratorio de Biotecnología y Biología Molecular, Departamento de Ciencias Básicas, Universidad Autónoma Agraria Antonio Narro, Calzada Antonio Narro 1923, Buenavista, Saltillo 25315, Coahuila, Mexico; jtafare@uaaan.edu.mx; 3CONACYT-Coordinación de Tecnología de Alimentos de Origen Vegetal, Centro de Investigación en Alimentación y Desarrollo A.C., Carretera Gustavo Enrique Astiazarán Rosas 46, Hermosillo 83304, Sonora, Mexico; miguel.hernandez@ciad.mx; 4Departamento de Investigación en Física, Universidad de Sonora, Blvd. Luis Encinas y Rosales S/N, Hermosillo 83000, Sonora, Mexico; alexel.burgara@unison.mx; 5Coordinación de Desarrollo Regional, Centro de Investigación en Alimentación y Desarrollo A.C., Carretera Gustavo Enrique Astiazarán Rosas 46, Hermosillo 83304, Sonora, Mexico; jrobles@ciad.mx

**Keywords:** fruit cuticle, cutin, wax, plant lipids, genes, cuticle composition, cuticle biosynthesis

## Abstract

Fleshy fruits represent a valuable resource of economic and nutritional relevance for humanity. The plant cuticle is the external lipid layer covering the nonwoody aerial organs of land plants, and it is the first contact between fruits and the environment. It has been hypothesized that the cuticle plays a role in the development, ripening, quality, resistance to pathogen attack and postharvest shelf life of fleshy fruits. The cuticle’s structure and composition change in response to the fruit’s developmental stage, fruit physiology and different postharvest treatments. This review summarizes current information on the physiology and molecular mechanism of cuticle biosynthesis and composition changes during the development, ripening and postharvest stages of fleshy fruits. A discussion and analysis of studies regarding the relationship between cuticle composition, water loss reduction and maintaining fleshy fruits’ postharvest quality are presented. An overview of the molecular mechanism of cuticle biosynthesis and efforts to elucidate it in fleshy fruits is included. Enhancing our knowledge about cuticle biosynthesis mechanisms and identifying specific transcripts, proteins and lipids related to quality traits in fleshy fruits could contribute to the design of biotechnological strategies to improve the quality and postharvest shelf life of these important fruit crops.

## 1. Introduction

Fleshy fruits are horticultural commodities with an edible and thick mesocarp or another tissue that are rich in water, sugars, fiber, minerals, vitamins, and antioxidant compounds. That is why fleshy fruits are a valuable source of nutrients for humanity—especially for lesser developed countries, where they are relatively accessible—and their production represents an important economic resource [1]. It has been predicted that global climate change will increase droughts in the following decades, reducing available water for agriculture [2]. In this scenario, it is necessary to develop biotechnological and agronomical strategies to develop and grow plants that are adapted to grow under conditions of limited water supply without affecting fruit productivity and quality. Because of that, it is necessary to carry out efforts to elucidate the molecular mechanism of drought adaptation and water loss avoidance in fleshy fruit-producing plants [2]. 

Despite fleshy fruits’ physical characteristics and the ontogeny among plant species being very diverse, the development and ripening of fleshy fruits occur through coordinated biochemical and molecular mechanisms, which appear to be evolutionarily conserved [3]. During fruit’s ripening, an increase in the production of antioxidant pigments, volatile compounds and sugars, accompanied by cell wall depolymerization and cuticle modification, occurs. Once the fruit has ripened and during postharvest, these changes increase the softening, water loss and susceptibility to pathogen infections [4], representing a major challenge for fleshy fruit commercialization and export.

The plant cuticle is the external lipidic layer covering land plants’ nonwoody aerial organs. The first land colonizer plants developed the ability to synthesize cuticles as a protection against desiccation, UV light exposure and extreme temperatures [5,6]. Because of this feature, distant evolutionary plants share some genes that play a role in fruit cuticle biosynthesis [7]. It is the first interaction between fruits and their surrounding environment and plays several roles in plant development, physiology and protection against biotic and abiotic stresses. One of its main functions is to protect the plant against water loss, acting as a transpiration barrier [8]. Thus, the cuticle has a relevant role in the development, ripening and postharvest shelf life of fleshy fruits [9,10].

Two main layers form the plant cuticle. The most internal layer, cutin, is covalently linked to the cell wall. It is formed mainly by long-chain fatty acids (LCFA) of 16 and 18 carbons and their oxygenated derivatives [5]. The most external layer, named cuticular wax, can be deposited over the cutin (epicuticular wax) or intercalated between the cutin components (intracuticular wax). Cuticular wax is formed by very long chain fatty acids (VLCFA) with 20 or more carbons in the main chain, and their derivatives, such as alkanes, alkenes, primary and secondary alcohols, aldehydes, ketones and esters, together with triterpenoids, steroids and phenolic compounds [11]. Generally, during organ growth, the cuticle increases in thickness. Nevertheless, specific cuticle deposition patterns, compositions and structures depend on plant species, organs, fruit cultivars, developmental stages and environmental conditions [6]. 

In fleshy fruits, as in other plant organs, cuticle composition is a dynamic characteristic that changes in response to developmental stages and environmental conditions [6]. Maintaining water content is essential for fleshy fruit development, and consequently, it has been hypothesized that the cuticle has a relevant function in developing and maintaining quality parameters during fruit development on plants [10]. Cuticle composition analysis during fruit development on plants and in postharvest conditions allows the elucidation of the association between the cuticle composition, quality parameters and longer shelf life of fleshy fruits. Several studies have been carried out that together suggest that the fleshy fruit cuticle plays a pivotal role in maintaining quality in postharvest shelf life, mainly by avoiding the fruit softening caused by water loss [8,12].

Due to the considerable importance of cuticles in plant physiology, development and defense against biotic and abiotic stress, efforts to elucidate the molecular pathway of cuticle biosynthesis have been carried out in plants—mainly in model plants such as Arabidopsis [13], tomato (*Solanum lycopersicum*) [14] and relevant crop plants [15]. Furthermore, it has been possible to identify orthologous genes and proteins playing a role in cuticle biosynthesis in other fruits through transcriptomic and proteomic tests. Excellent reviews have been published regarding the biosynthesis pathway and chemical composition of cuticles in plants [5,11] and fruits [16], their physiological functions [6] and their association with crop improvement [10,17].

This review describes the current information about cuticle biosynthesis and composition dynamics during growth, development and ripening and in the postharvest environment in fleshy fruits. An overview of studies regarding cuticle biosynthesis and composition in fleshy fruits and its relationship with water loss reduction and other relevant postharvest issues is presented. Gene expression analyses to identify transcripts and proteins playing a role in fleshy fruit cuticle biosynthesis are included and discussed. All this knowledge about the cuticle biosynthesis mechanism and identifying specific transcripts, proteins and metabolites related to quality traits in fleshy fruit-producing plants could contribute to developing biotechnological strategies to improve plant drought adaptation and reduce water loss for these economically and nutritionally important horticultural commodities.

## 2. Fleshy Fruit Cuticle Composition

Most studies carried out to analyze cuticle composition are based on chromatographic methods such as gas chromatography coupled to mass spectrometry (GC-MS), together with Fourier Transform Infrared Spectroscopy (FTIR) and Raman spectroscopy [16]. These tools provide valuable data to accurately compare tissues, species or developmental stages. Table 1 shows the main components of the cutin, intracuticular and epicuticular waxes of the cuticle from different fruit species.

Generally, a high proportion of alkanes and triterpenoids, followed by other very-long chain (VLC) aliphatic compounds, are found in the cuticular wax of fruits such as tomato, apple, sweet cherry, peach, pear, pepper and pitahaya [6]. Triterpenoids are the most prominent wax compounds in grape, olive and blueberries, whereas VLC aliphatics are in minor proportions [6]. In fruits such as orange, mandarin, lemon and jujube, VLC aliphatic waxes are found in significantly greater amounts than triterpenoids. 9(10),16-dihydroxy hexadecanoic acid is the main cutin component of tomato, apple, sweet cherry, drupe fruit, pepper, olive, guava and pitahaya. In fact, except for citrus fruits, 9(10),16-dihydroxy hexadecanoic acid is one of the most prevalent constituents of cutin in fleshy fruits. Nevertheless, the proportions of fruit cuticle components are very dynamic, and they vary during fruit development and ripening. 

### 2.1. Changes in Cuticle Composition during Fleshy Fruit Development and Ripening

Even though patterns of development and growing vary significantly among different types of fleshy fruits, they present a similar development and growing phenomena in which the first phase is characterized by many cycles of cell division and differentiation, with a low growth rate [4]. Subsequently, a phase of cell expansion begins, leading to a significant increase in fruit volume and mass during fruits’ development, which is characterized by the accumulation mainly of water and soluble solids [52,53]. Physiological maturity is attained when the fruit has reached its maximum size. After that state, the fruit ripening phenomenon is initiated [4]. 

The function of fleshy fruits is to protect the seed when the embryo has not been fully developed and promote fruit consumption and the dispersal of the seed once the embryo has been developed, so it is mature and ready to germinate and grow [1]. The depolymerization of the cell wall and changes in cuticle composition occur during fruit ripening. Once the fruit has ripened and during postharvest, these changes could increase softening, water loss and susceptibility to pathogen infections. Table 2 summarizes the cuticle composition changes recorded during fleshy fruit development phenomena. Next, studies carried out in mature fruits, during fruit development and ripening are summarized and discussed. 

#### 2.1.1. Tomato

In the case of the tomato fruit (*S. lycopersicum*) “Ailsa Craig” variety, the amount of wax of VLC aliphatic compounds increases continually throughout its development. During the developmental phenomena, the main components of cuticular waxes are alkanes and triterpenoids, but alkanes show the most significant increase. When compared with other species, an apparent particularity of tomato fruit waxes is the presence of large amounts of VLC unsaturated compounds (i.e., alkenes and alken-1-ols), which increase throughout fruit development and ripening [18]. 

Tomato fruit cuticle is mainly composed of cutin [20]. Tomato epicuticular and intracuticular waxes are mainly formed by VLC aliphatic compounds and pentacyclic triterpenoids, respectively [19]. Like waxes, cutin monomers increase throughout tomato fruit “Ailsa Craig” development and ripening. C_16_ fatty acids, including a large amount of 9(10),16-dihydroxy hexadecanoic acid, are the principal constituents (86–88%) of total cutin, whereas C_18_ fatty acids, with a large amount of 18-hydroxy octadecanoic acid, are the minor constituents (10–12%) [18]. 

The chemical composition of tomato fruit cuticle in relative wild-tomato species is very diverse and mainly varies in the occurrence of triterpenoid isomers and wax esters [20]. Like *S. lycopersicum* [18], the most prominent cutin constituent of tomato wild species *S. pimpinellifolium*, *S. cheesmaniae*, *S. chmielewskii*, *S. neorickii*, *S. habrochaites* and *S. pennellii* is 10,16-dihydroxy hexadecanoic acid. However, a notably significant amount of 9,10,18-trihydroxy octadecanoic acid is present in *S. chmielewskii*, *S. habrochaites* and *S. pennellii* species [20]. 

The main constituent of cuticular wax of all seven *Solanum* species IS alkanes, including as the most common and abundant nonacosane (C_29_) and hentriacontane (C_31_), but the total wax coverage of all the six wild species is higher compared with *S. lycopersicum.* The main non-aliphatic constituents identified were the pentacyclic triterpenoids amyrins. Nevertheless, triterpenoids represent a significant proportion of total wax composition only in *S. lycopersicum*, *S. pimpinellifolium* and *S. cheesmaniae* [20]. 

The tomato fruit mutants ripening inhibitor (*rin*), non-ripening (*nor*) and the landrace “Alcobaca”, which are characterized AS having a delayed or absent ripening process and substantially less softening, have higher amounts of wax and cutin than the typical tomato fruit ripening “Ailsa Craig”. A higher relative proportion of C_18_ monomers than C_16_ is shown in *rin*, *nor* and “Alcobaca” during development than in “Ailsa Craig”. It has been hypothesized that cutin with a large amount of C_16_ oxygenated fatty acids has a more rigid cuticle than cuticles constituted by an equal mixture of C_16_ and C_18_ oxygenated fatty acids in the cutin, which are more elastic. Furthermore, cuticle elasticity has been attributed to higher trihydroxy fatty acid content in cutin. Nevertheless, this hypothesis needs the support of more experimental data [18].

#### 2.1.2. Citrus

Very long chain fatty acids, alkanes and primary alcohols are the main constituents of the sweet orange cuticle (*Citrus sinensis* [L.] Osbeck) cultivar “Navelate” [22]. The epicuticular wax of the sweet orange fruit (*C. sinensis*) cutivar “Bingtang” is mainly composed of fatty acids, followed by alkanes. Intracuticular waxes are mainly composed of fatty acids, triterpenoids and alcohols. For cutin, the predominant constituents are *cis*-9-hexadecenoic acid and cinnamic acid [23]. In “Navelate” sweet orange, cuticle thickness slightly decreases during development. The total epicuticular wax load significantly increases only at the breaker (Bk) stage but remains unchanged at mature green (MG), colored (C) and full-colored (FC) stages. During sweet orange fruit development, VLC alkanes increase, whereas VLC fatty acids and VLC aldehyde proportion decrease [22].

Epicuticular waxes of the navel orange (*C. sinensis* [L.] Osbeck) cultivar “Newhall” are mainly composed of fatty acids, followed by alkanes and primary alcohols, whereas terpenoids are only present in low amounts. In contrast, intracuticular waxes are mainly composed of cyclic wax compounds, such as the triterpenoids, whereas aliphatic compounds are in low concentrations. In early navel orange fruit development stages, a low variety of epicuticular wax components is observed, but compound variety increases in the late stages. The main fatty acid constituents increase with further development [21].

The “Glossy Newhall” mutant displays a glossier peel than its wild-type navel orange (*C. sinensis* [L.] Osbeck cv. “Newhall”). During development, the total cuticular wax increases in navel orange and “glossy Newhall” with the same deposition pattern; nevertheless, “Glossy Newhall” presents a significantly lower wax load [56]. The main cuticular constituents of both navel orange and “Glossy Newhall” mature fruits are triterpenoids, followed by aldehydes, alkanes and fatty acids, but in “Glossy Newhall” mutant, aldehydes and alkanes are present in a lower amount [56].

During wild-type navel orange development, the total epicuticular wax increases from 60 to 120 days after flowering (DAF), decreases from 120 to 150 DAF and increases from 150 to 210 DAF. In contrast, in the “Glossy Newhall” mutant, the increase from 150 to 210 DAF does not occur [21]. These differences are consistent with the loss of wax crystals phenotype observed in “Glossy Newhall”, which have been associated with a decrease in the aldehydes and alkanes components of navel orange cuticle [21,56]. It has been suggested that cuticular crystals are mainly composed of VLC aliphatic constituents, especially by alkanes. 

Similar to navel orange “Newhall” [56], the cuticular wax of mandarin (*Citrus unshiu*) “Satsuma” mature fruit is mainly composed of aldehydes, alkanes, fatty acids and primary alcohols. Nevertheless, navel orange “Newhall” has a higher total cuticular wax and epicuticular wax amount than mandarin “Satsuma”. Furthermore, in cultivar “Newhall”, a significantly high amount of hentriacontane (C_31_), C_24_ and C_26_ chain length fatty acids and aldehydes are observed. In the case of terpenes, wax constituents, farnesol and squalene are only observed in navel oranges but not in mandarin fruit [24].

The main components in mandarin “Satsuma” fruit (*C. unshiu*) epicuticular wax are fatty acids, followed by alkanes and terpenoids. For intracuticular wax, the predominant constituents are terpenoids, followed by alkanes and fatty acids. The main constituents of cutin are cinnamic acids, followed by hexadecanedioic acid (C_16_) and hexadecanoic acid (C_16_) [25]. In mature green lemon fruits (*Citrus limon* Burm. f. Eureka), the main components of epicuticular wax are alkanes (C_23_-C_33_), with hentriacontane (C_31_) as the major component [26].

#### 2.1.3. Apple

At the completely ripe stage, the cuticular waxes of apple fruit (*Malus domestica* Borkh.) late-season “Florina” and early-season “Prima” cultivars are mainly composed of triterpenoids, which represent 70% and 83% of the total wax contents, respectively, of which ursolic acid and oleanolic acid are the most prominent. In “Florina”, the principal VLC aliphatic compounds are primary alcohols, whereas in “Prima”, these are secondary alcohols, fatty acids and alkanes. A slight but not significant difference in total wax amounts between the two cultivars is observed. Nevertheless, “Florina” presents a higher number of fatty acids, esters, primary alcohols and aldehydes than “Prima” cultivar fruits [29].

Cutins of “Florina” and “Prima” cultivars are mainly composed of hydroxylated hexadecanoic (C_16_) and octadecanoic (C_18_) acid monomers, of which 9(10),16-dihydroxy hexadecenoic acid is the most prominent, representing 34% of the total cutin content [29]. In both “Golden Delicious” and “Red Delicious” varieties, the main components of cutin are the fatty acids 9,10,18-trihydroxy octadecanoic (C_18_), 10,20-dihydroxycosanoic (C_20_), 10,16-dihydroxy hexadecanoic (C_16_), 9,10-epoxy-12-octadecenoic (C_18:1_) and 9,10-epoxy-18-hydroxy-12-octadecenoic acid (C_18:1_) [57]. In contrast to “Florina” and “Prima” cultivars, the main constituents of the cuticular wax of apple (*M. domestica* Borkh) cultivars “Red Delicious”, “Royal Gala”, “Granny Smith” and “Cripps Pink” are fatty acids, followed by alkanes and triterpenoids, with nonacosane (C_29_) and ursolic acid the main alkanes and triterpenoids compounds, respectively [58].

The main constituents of cuticular wax of the apple fruit (*M. domestica* Borkh.) cultivar “Starkrimson” at commercial maturity are fatty acids, primary alcohols and alkanes. Docosanoic (C_22_) and octacosanoic (C_28_) acids are the most prominent fatty acids, whereas pentacosanol (C_25_) and octacosanol (C_28_) are the most prominent primary alcohols [28]. The main components of “Red Fuji” apple (*M. domestica* Borkh.) cuticular wax are hydrocarbons, fatty acids, nonacosan-10-one and nonacosan-10-ol. Even-numbered straight-chains are the most prevalent fatty acids, whereas nonacosane (C_29_) is the major alkane compound. Both nonacosane (C_29_), and heptacosane (C_27_) amounts increase during “Red Fuji” apple fruit development [27].

For the apple cultivars “Stark”, “Golden”, “Mutsu”, “Golden Delicious”, “Camachi”, “Huahong”, “Jona Gold”, “Red Star”, “Ralls” and “Mashima Fuji”, the main wax constituents at harvest are alkanes, followed by fatty acids and primary alcohols. Terpenoids and aldehydes were only observed in the “Red Star” cultivar. Nevertheless, after 49 days of postharvest storage at room temperature and 90% relative humidity, terpenoids and aldehydes were found to be present in the 10 cultivars [59]. These analyses show that, like orange fruit, apples belonging to different cultivars can exhibit marked differences in cuticle composition at harvest and during the postharvest storage of apples.

#### 2.1.4. *Prunus* spp.

The main constituents of the cuticular wax of mature sweet cherry fruits (*Prunus avium*) cultivars “Rainier”, “Bing”, “Lapins”, “Kordia” and “Regina” are triterpenes, alkanes and alcohols with 76%, 19% and 1% of the total wax, respectively. Further, nonacosane (C_29_) is the most prominent alkane in all five cultivars [60]. Ursolic acid is the most abundant triterpene, accounting for 60% of total triterpenes. For alcohols, the secondary alcohol nonacosan-10-ol is the most abundant. As reported for tomato [18], mature sweet cherry fruit cutin is mainly composed of C_16_ monomers. Out of those, the most abundant is 9(10),16-dihydroxy hexadecanoic acid [30]. Wax and cutin accumulation increase during the first stages of development, but they markedly decrease at the latest developmental stages. During development, triterpenes decrease, but alkanes and alcohols remain practically unchanged. In the case of cutin, a decrease in the total mass of cutin during sweet cherry development due to a low deposition of the main cutin constituents was observed [30]. 

Cuticular waxes of cherry fruit (*P. avium* L.) cultivars “Celeste” and “Somerset” are mainly composed of ursolic acid, nonacosane (C_29_), linoleic acid and beta-sitosterol [61]. Unlike “Rainier”, “Bing”, “Lapins”, “Kordia” and “Regina” cultivars [30], cutin composition in “Celeste” and “Somerset” is mainly composed of C_18_ monomers. At commercial harvest, “Somerset” cherries are firmer, juicer, have higher quality indicators and have higher yields of cuticle than “Celeste” cherries. “Somerset” cuticles have large amounts of the cuticular wax phytosterols and alkanes, with sizes between C_27_ to C_31_ of chain length, and higher amounts of cutin than “Celeste”. Altogether, the above-mentioned data strongly suggest an association between cuticle yields and composition in maintaining fruit quality at harvest [61].

Nectarine (*Prunus persica* L. Batsch) cuticular waxes of “Summergrand” and “Zéphir” varieties are mainly composed of the triterpenoids oleanolic and ursolic acids. Minor compounds in nectarine wax are VLC aliphatics, mainly composed of alkanes, alcohols and fatty acids. Total waxes are low at the first stage of nectarine development, accumulate strongly at the middle stage, decrease at the end of the middle stage and slightly increase again near ripening. An early active cuticle accumulation process is observed during the first growth period, mainly due to triterpenoid deposition and cutin formation. On the other hand, the increase in waxes close to the final development and ripening stage consists mainly in alkane accumulation [32]. 

In mature peach fruits (*P. persica* L. Batsch.), the main wax constituents are the alkanes tricosane (C_23_) and pentacosane (C_25_) and the triterpenoids ursolic and oleanolic acids, while the main cutin constituents are the mono-unsaturated 18-hydroxyoleic acids (C_18:1_). For the “October Sun” cultivar, triterpenes, alkanes and fatty acids comprise 51.91%, 16.51%, and 8.27% of total wax constituents, respectively; whereas for the “Jesca” cultivar, these comprise 44.05%, 29.40% and 10.22% of total wax constituents, respectively. In “October Sun” and “Jesca” cultivars, C_18_ monomers represent 54.7 and 57.1% of total cutin constituents, respectively [31]. 

In agreement with that reported in other fruits of the genus *Prunus* [30,32,60], cuticular waxes of drupe fruit (*Prunus laurocerasus* L.) are mainly composed of pentacyclic triterpenoids, which account for 87% of total waxes, with ursolic acid the most prominent component. The aliphatic wax compounds are mainly composed of fatty acids, alkanes and primary alcohols, with about 10% of total wax. Nonacosane (C_29_) and triacontanoic acid (C_30_) are the main aliphatic compounds during drupe fruit development. The cutin is mainly made of 9(10),-dihydroxy hexadecanoic acid, 9,10-epoxy-18-hydroxy octadecanoic acid and 9,10,18-trihydroxy octadecanoic acid, comprising more than 70% of the total cutin monomers. At later stages of development, drupe shows higher amounts of total cutin and triterpenoids and a higher abundance of 3,4-dihydroxy cinnamic acid than at earlier stages [33].

#### 2.1.5. Pear

The epicuticular wax of mature pear (*Pyrus* spp.) fruits from the species *P. communis* Linn., *P. ussuriensis* Maxim., *P. sinkiangensis* Yü., *P. bretschneideri* Rehd. and *P. pyrifolia* Burm Nakai is mainly composed of alkanes, primary alcohols and terpenoids, with 40.72%, 24.47% and 11.8% of the total content, respectively. Alkanes of 28 to 31 carbon chain lengths are the most abundant, while the alcohol triacontanol (C_30_) is the most abundant [37]. The cuticular wax of Asian pear cultivars “Kuerle”, “Xuehua” and “Yuluxiang” are very similar and are mainly composed of alkanes and primary alcohols. The “Kuerle” cultivar shows a higher cuticular wax amount than “Xuehua” and “Yuluxiang”, with “Xuehua” the cultivar with the lowest amount. The most abundant wax fractions of “Kuerle” and “Yuluxiang” are alkanes and primary alcohols, whereas in the case of “Xuehua”, these are primary alcohols and terpenoids [36].

The cuticular wax of wild-type “Dangshansuli” pear fruit (*Pyrus bretschneideri*) and its russet skin mutant “Xiusu” is mainly composed of alkanes, alkenes, fatty acids, alcohols and terpenes. Alkanes are the main compounds during pear fruit growth and development, with nanocosane (C_29_) as the major component. There was no difference between alkane content in both “Dangshansuli” and “Xiusu” cultivars at early stages of development. Nevertheless, a higher content of alkanes is synthesized in “Dangshansuli” fruits during growth. Nonacosane (C_29_) content increases significantly in “Dangshansuli” fruits during ripening, becoming the most abundant alkane compound during “Dangshansuli” pear development [35].

Fatty acid content, mainly composed of hexadecanoic and octadecanoic acid, is higher during the early stages of “Dangshansuli” and “Xiusu” pear fruit development, decreasing in the middle and late stages. In contrast, for triterpenoids, mainly composed of alpha-glycosidal and beta-glycosidal, a continuous increase in relative content is shown during all development stages of both cultivars. Fatty alcohol content is higher in “Xiusu” pear than in “Dangshansuli”, and this difference is more obvious during the latest stage of the ripening phenomena. A gradual increase in cuticle thickness is present in “Dangshansuli” fruit, whereas no clear deposition pattern is observed for “Xiusu” fruit [35]. 

Similarly, the main constituents of cuticular wax of Asian pear fruit (*P. bretchneideri* Rehd) cultivar “Pingguoli” are alkanes, fatty acids and triterpenoids, accounting for 25.9%, 27.8% and 33.6% of the total waxes, respectively. Straight-chain odd-numbered compounds are the main alkane components, and out of these, nonacosane (C_29_) and heptacosane (C_27_) are the most prominent. Fatty acids include saturated and unsaturated straight-chain odd and even-numbered compounds. In contrast, for triterpenoids, alpha-amyrin is the most abundant [34]. Changes in total wax content, relative amount and carbon chain length have been observed during “Pingguoli” pear fruit development. In earlier stages, alkanes and fatty acids are the main wax compounds, whereas alkanes and triterpenoids predominate in later stages. Total wax content increases abruptly in later stages of development, while a reduction in alkane proportion is shown during development [62].

#### 2.1.6. Berries (*Vaccinium* spp.)

At the ripe stage, the cuticular wax of berries of the *Vaccinium* genus is mainly composed of triterpenoids, but significant variations in composition have been shown between berries belonging to different species [63] and among cultivars [38]. Through the analysis of nine blueberry cultivars belonging to the species *Vaccinium corymbosum* (cv. “Misty”, “O’Neal”, “Sharpblue”, “Brigitta”, “Darrow” and “Legacy”) and *V. ashei* (cv. “Britewell”, “Premier” and “Powderblue”), it was shown that their cuticular waxes are mainly composed of triterpenoids and beta-diketones, accounting for 64.2% and 16.4% of the total wax, respectively. Nevertheless, the total wax composition differs considerably between cultivars. Hentriacontan-10,12-dione was only detected in *V. corymbosum*, whereas tritriacontan-12,14-dione was only detected in *V. ashei*. Except for beta-diketones, VLC aliphatic compounds are minor constituents of cuticular wax. For aldehydes, fatty acids and primary alcohols, the main constituents are C_28_ and C_30_ compounds, whereas for alkanes, the main constituent is C_29_ [39].

The cuticular wax of blueberry cultivars “Legacy” (*V. corymbosum*) and “Brightwell” (*V. ashei*) is composed mainly of oleanolic and ursolic acid, followed by hentriacontane-10,12-dione and tritriacontane-12,14-dione, respectively. During ripening, the total wax amount and the triterpenoid content increase continuously for both cultivars; nevertheless, “Brightwell” shows a higher wax content than “Legacy”. A decrease in the relative content of diketones occurs during the ripening of both blueberry cultivars with an increase in VLC aldehydes, primary alcohols, fatty acids and alkanes [40]. At ripening stages, the triterpenoid wax fraction of blueberry (*V. corymbosum* L.) cultivars “Brigitta” and “Duke” is mainly composed of lupeol, oleanolic acid and ursolic acid, whereas alpha-amyrin is only observed in the “Brigitta” cultivar [38]. Altogether, these studies showed a cultivar-dependent cuticular wax deposition and composition for blueberry fruits.

The most prominent components of lingonberry (*V. vitis-idaea* L.) and bog bilberry (*V. uliginosum* L.) waxes are triterpenoids and fatty acids, respectively, representing more than 50% of the total wax. In contrast, for bilberry (*V. myrtillus* L.), the triterpenoids and fatty acids mixture comprises more than 70% of the cuticular wax composition [41]. In the first stages of development, when berries are in stages of intensive growth, steroids are in very low amounts in the cuticle, but they increase during the stage of maturation once the berries have reached their final size. A species-specific pattern of wax composition has been observed during berries’ fruit development, with similarities among phylogenetically related species [64]. 

Both wild-type bilberry (*V. myrtillus*) and its “glossy mutant” GT have similar patterns of cuticular wax deposition and composition through development, consisting of a decrease in triterpenes and an increase in aliphatic compounds. The dominant compounds during bilberries’ development are triterpenoids and fatty acids. Despite no significant differences in total wax amount being detected between wild-type and GT during development, the glaucous phenotype could be due to a higher proportion of triterpenes and a lower proportion of fatty acids and ketones present in bilberry GT [54].

#### 2.1.7. Grape

Cuticular waxes of grapes (*Vitis vinifera* L.) varieties “Müller Thurgau” and “Blauer Spätburgunder” are mainly composed of oleanolic acid, but this compound is absent in epicuticular wax. Triterpenoid deposition begins and increases early in grapevine fruit development and decreases during ripening. Except for alkyl esters and fatty acids, whose deposition does not decrease during ripening, VLC aliphatic compounds have the same pattern of deposition to triterpenoids, but they are present in much smaller amounts [42]. Like oranges [21,56], the epicuticular wax crystals of grapevine fruit are formed apparently only by VLC aliphatic compounds. The main components of epicuticular wax are alcohols, which are present in higher amounts in the first stages of development and decrease gradually during grapevine development. In contrast, alkyl esters and fatty acids are in much smaller amounts in the first stages, but they increase in the last stage of development [42]. 

The triterpenoid wax composition of the eight grape (*V. vinifera*) cultivars “Chasselas”, “Gewurztraminer”, “Muscat d’Alsace”, “Pinot auxerrois”, “Pinot gris”, “Pinot noir”, “Riesling” and “Sylvaner” are quite similar. Nevertheless, they exhibit differences during the grapefruit ripening phenomena. Oleanolic acid and oleanolic aldehyde are the most prominent constituents of the grape cuticle triterpenoid fraction for all eight cultivars. In earlier stages of ripening, a high level of total triterpenoids is observed, with a gradual decrease with the progression in development. “Gewurztraminer” shows an exceptionally higher amount of sitosterol than other cultivars, whereas 3,12-oleandione was only present in the “Muscat d’Alsace” cultivar [43].

The total aliphatic amount is highest at the early stages and slightly decreases at the end of grape berry “Gewürztraminer” cultivar development. Total triterpenoids, VLC aldehydes and VLC primary alcohols show a similar pattern of deposition to the total aliphatic amount, but their decrease is more obvious at the end of development. In contrast, VLC fatty acids and VLC esters increase during development. Alkanes are minor constituents of the cuticular wax of berries and show no significant changes during grape berry development [44].

Both total cuticular wax and epicuticular wax of mature grape berry (*V. vinifera*) cultivars “Kyoho”, “Muscat Hamburg”, “Redglobe” and “Zuijinxiang” are mainly composed of terpenoids, alcohols, fatty acids and esters. In agreement with the findings reported for other grape cultivars [42,43], the most abundant terpenoid among the four cultivars analyzed in this study is oleanolic acid. Grape berry cuticular terpenoids are mainly present in intracuticular wax, fatty acids are mainly present in epicuticular wax, whereas hydrocarbons are equally distributed in intra and epicuticular wax [45].

#### 2.1.8. Pepper

A study carried out in 50 diverse pepper (*Capsicum* spp.) accessions, including the genera *C. annuum*, *C. chinense*, *C. baccatum*, *C. pubescens* and *C. frutescens*, recorded that the main wax constituents are alkanes and nonaliphatic compounds including triterpenoids and phytosterols. The amount of alkanes ranges from 13% to 74% of the total waxes, with nonacosane (C_29_) and hentriacontane (C_31_) the most abundant alkanes. Nonaliphatic compounds range from 10% to 76%, with the main components being alpha and beta-amyrin, glutinol and lupeol. Pepper cutin comprises C_16_ and C_18_ fatty acids and their oxygenated derivatives, p-coumaric and m-coumaric acids. Similar to as observed for tomato [18] and sweet cherry [30], the main cutin constituents of pepper fruit are C_16_ monomers, ranging from 54% to 87%. Among those, the most important is 9(10),16-dihydroxy hexadecanoic acid, ranging from 50% to 82% of total cutin [46,65].

#### 2.1.9. Olive

The cuticular wax of the olive (*Olea europaea*) cultivar “Arbequina” is mainly composed of triterpenoids (74–62%), followed by primary alcohols (9–11%) and fatty acids (8–9%). Further, among the most abundant components are oleanolic acid, hexacosanol (C_26_) and hexacosanoic acid (C_26_), respectively. No changes have been observed in total wax content and triterpenoid content during olive development; nevertheless, the VLC acyclic compounds increase. Despite no differences in total cutin amount being observed between the developmental stages of olive fruit, fatty acids, ω-hydroxy fatty acids and ω-hydroxy fatty acids with mid-chain hydroxy groups increased. Out of these, the predominant cutin monomers are 9(10),16-dihydroxy hexadecanoic, 9,10,18-trihydroxy octadecenoic and 9,10,18-trihydroxy octadecanoic acid [47].

#### 2.1.10. Guava

The cuticular wax of the physiologically mature guava fruit (*Psidium guajava* L.) cultivar “Pearl” is mainly composed of fatty acids, primary alcohols and triterpenoids, such as uvaol, ursolic and maslinic acid, which are the most abundant. Octacosanoic acid (C_28_) and triacontanol (C_30_) are the most prominent constituents of fatty acids and primary alcohols, respectively. In cutin, the principal constituents are the monomers 9(10),16-dihydroxy hexadecanoic acid and 9,10-epoxy-18–hydroxy octadecanoic acid [48].

#### 2.1.11. Pitahaya

At the mature stage, the wax of the pitahaya fruit (*Hylocereus polyrhizus*) cultivar “Hongshuijing” mainly comprises triterpenoids, followed by alkanes and fatty acids. Alkanes range from C_20_ to C_35_ of chain length, dominated by hentriacontane (C_31_) and tritiacontane (C_33_), whereas the most prominent triterpenoids are uvaol, lupenon, beta-amyrinon and beta-amyrin. Cutin comprises 9(10),16-dihydroxy hexadecanoic acid and 9,10-epoxy-18-hydroxy octadecanoic acid. The cuticle of pitahaya presents a wax/cutin ratio of 0.6 and compounds with an average chain length (ACL) of 30.5. Authors argue that ACL is higher than that reported for other petal and fruit cuticles and similar to that reported for leaf cuticles of other species [49]. 

One of the most abundant components of fleshy fruits is water, which is essential for fruit metabolism, fruit development and size increase [53]. In fleshy fruits, most water loss occurs through transpiration through the fruit surface [53], which is covered by the cuticle. It has been suggested that high wax/cutin ratios and compounds with high ACL could be a physiological adaptation to enhance the transpiration barrier of the fruit cuticle to withstand arid environments [49]. In addition, a high triterpenoids content could strengthen the mechanical support and plasticity of the cuticular membrane, protecting the fruit cuticle from the harmful effects of high-temperature stress [49]. In the next section of this review, studies regarding the physiological function of cuticles at harvest and during postharvest in fleshy fruits are described and discussed, mainly regarding fruit quality maintenance and water loss reduction.

## 3. Physiological Function of Fleshy Fruit Cuticle

### 3.1. Water Loss

The loss-of-function mutation of beta-ketoacyl-CoA synthase (KCS) in tomato fruit, designated as *SlCER*6 mutant, causes a decrease of alkanes and aldehydes longer than C_30_ and an increase of intracuticular triterpenoids in the cuticle. An increase in permeability has been observed due to the reduction of the intracuticular VLC aliphatic compounds. The authors suggest that the transpiration barrier is mainly determined by the proportions of VLC aliphatic constituents and triterpenoids of the intracuticular waxes rather than epicuticular wax’s VLC aliphatic composition [19].

During tomato fruit development, *SlCER*6 shows a higher cuticle amount than wild type “MicroTom”. Nevertheless, a reduction in VLC alkanes and an increase in cyclic triterpenoids, together with an increase in water loss, are shown in waxes of *SlCER6* [66]. Positional sterile (*ps*) mutant of tomato exhibits a similar phenotype to *SlCER6*, which is defective in the elongation of VLC fatty acids (VLCFA). No difference in cutin composition, cutin accumulation and cuticular wax accumulation between *the mutant ps* and its wild type is observed. Nevertheless, the wild type exhibits a remarkably higher level of alkanes and aldehydes in cuticular waxes and a remarkably lower level of esters and triterpenoids than *ps* fruits, with hentriacontane (C_31_) the most prominent alkane constituent in wild-type tomato waxes. Furthermore, a reduction in growth and weight and an increase in water loss are observed in *ps* fruits [67]. 

Studies with tomato fruit strongly support the statements that (i) cuticle rather than cell wall modifications could have a significant role in the reduction of the softening rate in fleshy fruits [12], (ii) waxes are the main cuticular structures that regulate the permeability in fleshy fruit peels [19,66], and that (iii) a specific change in cuticle wax composition rather than an increase in thickness or the total amount of cuticle has a more significant effect on the properties of fleshy fruits [19,66], especially for the reduction of water rate loss. Furthermore, several studies showed that epicuticular waxes, rather than intracuticular waxes, have a significant effect on fleshy fruit cuticle permeability [45], although other studies have shown opposite results [19]. Therefore, more experimental evidence is needed to fully elucidate the role of epicuticular and intracuticular waxes in the permeability properties of the cuticle. 

In “Navelate” sweet orange, the total epicuticular wax load significantly increases in the earlier stage of development, breaker (Bk), but remains unchanged in the later stages, namely mature green (MG), colored (C) and full-colored (FC). Terpenoids are mainly present in the Bk stage, and a notorious increase of hentriacontane (C_31_) happens in this stage. Cuticle transpiration and permeability are lower in the first two stages of development (Bk and MG), suggesting that alkanes and triterpenoids have a relevant function in controlling cuticle permeability during sweet orange fruit development [22].

It had been suggested that the early rise of epicuticular crystals, alcohols and triterpenoids observed at the beginning of the development of grapevine fruit could be due to a mechanism to protect the underlying cuticle from the exponential increase in volume and rapid surface expansion [42]. Like sweet orange [22] and grapevine [42], during nectarine development, an early active cuticle accumulation process is observed during the first growth period, mainly by triterpenoid deposition and cutin formation. Authors suggest that the increase in wax accumulation during the early stages of development allows the fruit to reduce water loss at harvest and during postharvest conditions [32]. 

The removal of epicuticular waxes in grape berry induces higher water loss and softening, indicating that epicuticular waxes could play a pivotal role in the postharvest quality of grape berry by reducing the water loss rate [45]. 

During postharvest storage, alkanes and aldehydes of Korla pear fruit cuticular waxes negatively correlate with weight loss, whereas fatty acids and alcohols have a positive correlation. The composition and morphological analysis of the Korla pear cuticle suggests that alkanes and aldehydes could contribute to wax crystal formation and consequently fruit water retention in Korla pear during postharvest storage [68].

Water deficit induces total aliphatic waxes accumulation at the end of grape berry development, accompanied by an obvious increase of VLC esters. This increase in cuticular VLC esters could be due to an adaptation to low water availability in grape berries. Nevertheless, these changes of composition do not cause a significant change in the berry transpiration rate [44]. On the other hand, it has been observed that water deficit leads to a lower triterpenoids/total aliphatic wax ratio in green and red berries. 

The apple fruit (*M. domestica*) cultivar “Florina” shows higher amounts of fatty acids, esters, primary alcohols and aldehydes than the “Prima” cultivar. However, it was not possible to establish a relationship between water permeability and cuticle composition for both cultivars [29].

It has been reported that the cuticle of physiologically mature guava fruit (*P. guajava* L.) has a large abundance of cyclic components, epoxy, hydroxy and carboxyl functional groups, and a relatively smaller amount of ACL of acyclic components than cuticles of other species and organs, which has been related with the low ability to reduce the transpiration of guava fruit cuticle [48]. In agreement, the cuticle of mature pitahaya fruit (*H. polyrhizus*) presents a wax/cutin ratio of 0.6 and an ACL of 30.5. Authors argue that the ACL is higher than that reported for other petal and fruit cuticles, similar to that reported for leaf cuticles of other plant species. They attribute these changes to the enhancement of the transpiration barrier of the pitahaya cuticle to withstand arid environments [49].

Cuticle composition analyses of pepper species *C. annuum* and *C. chinense*, which have high and low postharvest water loss rates, respectively, showed that postharvest fruit water loss is associated with the cuticle composition and the ratio of the wax constituents rather than the total wax amount. It has been shown that an increase in the total amounts of triterpenoid and sterols in the cuticular waxes of pepper fruits could increase postharvest water loss, while an increase in the amounts of primary alcohols and alkanes could reduce it—specifically, an increase of C_29_ and C_31_ alkanes [46]. 

In the case of cutin, it has been shown that water loss has a negative association with the C_16_/C_18_ ratio. The increase of total C_16_ monomers and 9(10),16-dihydroxy hexadecenoic acid in the cutin appears to induce postharvest fruit water loss in pepper fruits [46]. During olive development, a slight increase in the ACL of VLC acyclic compounds was observed, with a slight reduction of the C_16_/C_18_ ratio of cutin monomers; nevertheless, no differences were observed in cuticular permeability and water loss during olive fruit development [47].

The tomato cultivar “Delayed Fruit Deterioration” (DFD) exhibits a remarkable delayed softening at postharvest and remains firm for at least six months at fully ripe stages, but it otherwise undergoes a normal ripening process. No significant differences were found in the patterns of wall polysaccharide modification and the expression of genes related to wall degradation between DFD and the normal softening AC cultivar. However, DFD showed a lower transpiration rate, lower water loss, higher cellular turgor, higher firmness and a thicker and more swelled pericarp at ripening stages than AC [12].

Although DFD and AC have similar cuticle anatomies, DFD shows a stronger cuticle and a higher total wax amount. Besides, a significant increase in alkadienes in red ripe fruits is shown in DFD, but not in AC. Based on these data, the authors suggested that the delay of the softening phenotype could be due to the significant increase in alkadienes and the absence of naringenin in the cuticular wax of DFD, which has been associated with both mechanical support and turgor maintenance through water loss reduction [12]. In agreement, it has been shown that water stress increases fruit firmness and total cuticle, total wax and triterpenoids amounts, whereas it decreases cuticle permeability, transpiration rate and the relative amount of VLC alkanes in AC tomato fruits, which suggests an association between cuticle characteristics, transpiration and fruit firmness. In addition, an increase in total cutin amounts and 9(10),16-dihydroxy hexadecenoic acid was shown in AC [9].

Water stress does not affect cuticle permeability and thickness in DFD but induces an increase in both characteristics in AC tomato fruits. Actually, after water stress, the cuticle of AC shows similar characteristics to that of DFD, which suggests a change in cuticle metabolism in response to low water availability conditions in tomato [9]. It has been suggested that high amounts of alkanes and low amounts of triterpenoids in cuticles reduce the transpiration of fruit surfaces in fleshy fruits [46,66,67]. This study showed that cuticle permeability was positive and negatively correlated with the total cutin amount and the proportion of VLC alkanes in AC fruits, respectively. Besides, the levels of cyclic triterpenoids were positively correlated with the water loss rate [9]. 

The cuticles of mango cultivars “Kent”, “Tommy Atkins”, “Manila”, “Ataulfo”, “Criollo” and “Manila” exhibit different epicuticular wax deposition patterns, architectures and cutin compositions at the mature-green stage and during postharvest, accompanied by different water transpiration rates, firmness and fruit quality appearance. During postharvest, the mango “Tommy” cultivar has a higher wax deposition and cuticle thickness and exhibits a lower percentage of weight loss and less visual deterioration than the “Criollo” cultivar. Authors suggest that cuticle characteristics observed in premium cultivars such as “Tommy” are potential factors that could be associated with fruit quality preservation during postharvest storage [69]. 

### 3.2. Postharvest Storage

The amounts of both epi- and intracuticular waxes of mandarin “Satsuma” fruit (*C. unshiu*) increase after 20 days of room temperature (25 °C) storage, but they decrease after 40 days. A decrease in terpenoids and fatty acids and an increase in the proportion of alkanes is shown after 40 days of storage. Further, the total cutin amount decreases during postharvest storage, but the proportion of almost all cutin components remains stable [25]. The navel orange cultivar “Newhall” has a higher total cuticular wax and epicuticular wax amount than mandarin “Satsuma”. Notably, a significantly higher amount of hentriacontane (C_31_), and C_24_ and C_26_ chain length fatty acids and aldehydes is observed. During seven days at 25 °C and 40–50% relative humidity of postharvest conditions, navel orange exhibited a lower weight loss than mandarin [24], which could be due to the difference in the epicuticular wax content and composition observed.

During postharvest, the peach melting cultivar “October Sun” shows a more dramatic firmness loss and weight loss than the non-melting cultivar “Jesca”. At harvest, wax percentages are similar in both cultivars, whereas cutin percentages are significantly higher in “October Sun” than in “Jesca”. Five days after harvest, the total wax and cutin yields remain unchanged in “October Sun”, whereas in the “Jesca” cultivar, both wax and cutin significantly increase. At commercial harvest, the ratio of alkanes to triterpenoids and sterols is 0.31 in “October Sun” and 0.65 in “Jesca”. These data strongly suggest that the increasing alkanes play a role in maintaining the firmness and reducing weight loss in non-melting “Jesca” cultivars [31].

The total wax content of Korla pear fruit increases during 30 days of postharvest storage, but it decreases at day 90. Furthermore, alkanes and aldehydes show a negative correlation with weight loss of Korla pear during postharvest, whereas fatty acids and alcohols have a positive correlation [68], suggesting that alkanes play an important role in reducing water loss. In apple cultivars “Stark”, “Golden”, “Mutsu”, “Golden Delicious”, “Camachi”, “Huahong”, “Jona Gold”, “Red Star”, “Ralls” and “Mashima Fuji”, a decrease in total cuticular wax amount is observed after 49 days of postharvest storage, with a decrease in alkanes and primary alcohols and an increase in fatty acids proportion. A relationship between weight loss rate and total wax, total alkanes and C_54_ alkanes is shown in all the 10 cultivars, suggesting that alkane biosynthesis is essential for reducing weight loss during postharvest storage in apples [59].

Through the cuticle wax analysis of 35 cultivars of pear (*Pyrus* spp.) mature fruits, it has been shown that the cultivar with the longest postharvest storage period also showed a higher wax concentration [37]. In berries, lingonberry fruit (*V. vitis-idaea*), which is characterized as having a longer shelf-life than honeysuckle (*Lonicera caerulea*), and strawberry tree (*Arbutus unedo*), a higher content of triterpenoid acids in the cuticle was recorded. It has been suggested that triterpenoid acids might be related to lingonberry surface firmness and durability, probably due to the mechanical properties that they provide and the antimicrobial effect [64].

The main quality preservation strategies used to maintain the fruit quality at postharvest are based on temperature regulation such as cold storage and the application of ethylene regulators such as 1-methyl cyclopropane (1-MCP). Furthermore, controlled atmospheres are utilized with the same goal [70]. Nevertheless, during development and postharvest, fleshy fruits are susceptible to phytopathogen attack and the development of physiological disorders, such as cracking, russeting and chilling injury [70]. In Table 3, cuticle composition changes observed in response to postharvest storage conditions in fruits are shown. It has been reported that cuticle composition could have a relevant function regarding phytopathogens and the susceptibility to physical disorders. In the next section, the effect of postharvest treatments on cuticle composition and the relationship between cuticle composition, phytopathogen attack and physical disorders susceptibility are described. 

### 3.3. Cold Storage

Different patterns of changes in response to cold storage have been shown among apple cultivars. The main constituents of the cuticular wax of apple “Maxi Gala” fruit after nine months of cold storage are alkanes and fatty acids, with nonacosane (C_29_) and *cis*-13,16-docosadienoic acid (C_22:2_) the main compounds of these two fractions, respectively [72]. The amounts of total cuticular wax and the main alkane constituents nonacosane (C_29_) and heptacosane (C_27_) decrease during seven months of postharvest storage at 0 °C in “Red Fuji” apple fruit [27]. During 140 days of cold storage, the total wax content of apple fruit cultivar “Starkrimson” increases from day 0 to day 80, then decreases at day 140 [28].

The total epicuticular wax content of sweet orange fruit increases after 30 days of postharvest cold storage (4 °C), then decreases at day 40, whereas at 25 °C, a continuous increase occurs during 40 days of storage. At 4 °C, the total cutin amount decreases continuously, whereas at 25 °C, an increase is observed at 20 and 40 days of storage. At 4 °C, triterpenoids increase continuously during 20 days and then decrease after 40 days of storage. At the same time, a continuous increase in triterpenoids and a decrease in fatty acid is observed during 40 days of storage at 25 °C. At 4 °C, alkane composition remains stable, whereas at 25 °C, the alkane fraction increases. Moreover, nonacosane (C_29_) becomes the main alkane after 40 days of storage at 25 °C [23].

Changes in cuticle amounts and composition have been observed in response to postharvest at 20 °C and 0 °C for both “Somerset” and “Celeste” sweet cherry fruit cultivars, with a general increase in cuticle amount in response to cold storage [61]. Ursolic acid content has been positively associated with weight loss and softening of blueberry fruit at postharvest cold storage, whereas a negative association has been reported for oleanolic acid. During 45 days of postharvest storage at 0 °C, blueberry “Duke” was more prone to softening and dehydration than the “Brigitta” variety, which was highly correlated with the higher ursolic acid content in the triterpenoid wax fraction of “Duke” blueberry [38]. 

Cold storage at 4 °C for 30 days reduces the total wax content of both “Legacy” (*V. corymbosum*) and “Brightwell” (*V. ashei*) blueberry varieties, but differences in wax composition between both cultivars in response to cold storage have been reported. For the “Legacy” cultivar, diketones are the only VLC compound that decrease during the storage at 4 °C, whereas for “Brightwell”, a decrease in the content of all aliphatic VLC compounds was observed [40].

### 3.4. Heat and UV Light

The quality of fruits is affected by excessive exposure to heat and UV light, mainly due to the oxidation of proteins and enzymes. One of the physiological functions of fruit cuticles is protecting against UV light exposure and extreme temperatures [5,6]. An increase in thickness, cinnamic acid derivatives and chalconaringenin compounds of cuticle appears to play a pivotal role in modulating UV radiation exposure in tomato fruit [73]. Furthermore, conformational changes leading to the glass transition of the cuticle membrane could serve as an adaptation mechanism in response to a change in environmental conditions [73].

It has been shown that the heat capacity of cuticle depends on the developmental stages of tomato fruit and that the thermal properties of fruit cuticle could be regulated by phenolic compounds [73]. Heat treatment increases the wax contents amount of “October Sun” peach fruits, whereas the effect of heat on cutin is less clear [74]. After a room temperature storage period, peach subjected to heat treatment and cold storage showed a reduction of cutin amount. Furthermore, it has been shown that heat treatment reduces the acyclic/cyclic compounds ratio of peach fruits [74], which strongly suggests a major role of wax cyclic compounds in response to heat.

### 3.5. Ethylene Regulators and Controlled Atmosphere

A controlled atmosphere (CA), dynamic controlled atmosphere (DCA) based on chlorophyll fluorescence (DCA-CF), DCA respiratory quotient (DCA-RQ) and 1-MCP application to apple fruits “Maxi Gala” do not affect the total cuticular wax content during cold storage conditions. Nevertheless, these treatments induce a change in the composition or concentration of specific wax constituents. DCA-CF leads to a higher concentration of nonacosane (C_29_) and a reduction of mass loss, whereas 1-MCP reduces the concentration of nonacosan-10-ol and specific fatty acid constituents [72]. Storage of mature apple “Cripps Pink” under CA, DCA-CF and DCA-RQ treatments increases the wax concentration from day 7 to 14 of shelf life at 20 °C. Besides, all treatments led to a general increase of unsaturated fatty acids. Particularly, an increase in *cis*-11,14-eicosadienoic acid (C_20:2_), nonacosane (C_29_) and tetracosanal (C_24_) was observed. A controlled atmosphere leads to an increase in ursolic and oleanolic acids, whereas DCA-RQ leads to an increase in 10-nonacosanol (C_29_) [71].

In apple fruit cv. “Starkrimson”, the combination of ethephon treatment and storage at 0 °C accelerates total wax and VLC aliphatic deposition, whereas 1-MCP causes the opposite effect. One of the most obvious effects of ethephon and 1-MCP treatments was the increasing and decreasing of octacosanoic acid content, respectively [28]. In “Red Fuji” apple fruit, the total cuticular wax, nonacosane and heptacosane amounts decrease during seven months of postharvest storage at 0 °C, but 1-MCP treatment slightly suppresses this reduction [27]. Like “Starkrimson” apple [28], in “Red Fuji” apple, nonacosan-10-ol, nonacosan-10-one and hexadecanoic acid amounts increased after seven months of cold storage, but when fruits were treated with 1-MCP, their amounts were reduced [27]. Altogether, these findings support the relevant role of ethylene on the regulation of cuticular wax biosynthesis during the postharvest storage of fleshy fruits.

### 3.6. Physiological Disorders

The specific aliphatic composition of cutin influences the mechanical properties of apple fruit cuticles. It has been proposed that microcrack formation is due to low elastic cutin properties due to the low presence of phenolic compounds [57]. Sweet cherry varieties with a higher amount of nonacosane (C_29_) are more tolerant to cracking than those with lower levels, suggesting that this wax component could protect sweet cherry from cracking [60]. There were no differences between alkane contents in both wild-type “Dangshansuli” pear fruit and its russet mutant “Xiusu” at early stages of development. Nevertheless, as the fruit grows, a higher content of alkanes is synthesized in “Dangshansuli” fruits. These differences in wax deposition and composition during pear development could contribute to russeting formation observed in the “Xiusu” fruit [35].

The cuticular waxes of jujube fruit (*Ziziphus jujuba* Mill.) cultivars “Popozao”, “Banzao” and “Hupingzao” are mainly composed of fatty acids, primary alcohols and alkanes. No significant differences were observed in the mass of cuticle or cutin between the cracking-resistant cultivar “Popozao” and the cracking-susceptible cultivar “Hupingzao”. Nevertheless, in the coloring stage of jujube development, “Popozao” shows a higher level of total wax than “Hupingzao”. It has been suggested that the severity of microcracks during fruit development could be related to a lower level of cuticular wax. Furthermore, during the coloring period, “Popozao” cuticular wax contains fewer fatty acids but more alkanes and aldehydes with a chain length greater than 20 carbon atoms than cultivars “Banzao” and “Hupingzao”. Based on the above-mentioned factors, it seems that alkanes and aldehydes with longer chain lengths could contribute to protecting against microcracking during the coloring period of jujube fruit enlargement [50]. 

No difference in thickness and total content of epicuticular wax is shown in oleocellosis-damaged lemon fruits, but a significant increase of alkanes (especially C_29_) and a decrease of the amount of aldehydes (especially C_32_) are shown, suggesting that oleocellosis can be related to the transformation of VLC aldehydes to VLC alkanes [26]. In Table 4, a summary of changes in cuticle composition reported in response to the appearance of skin disorders in fleshy fruits is shown.

### 3.7. Pathogen Infection

In Asian pear, a negative association between cuticular wax concentration and the development of Alternaria rot has been shown, with a difference in resistance to Alternaria rot between cultivars [36]. Cuticular waxes of mature goji berry fruit (*L. barbarum* L.) are mainly composed of fatty acids, alkanes and primary alcohols, accounting for 47.09%, 21.66% and 11.98% of the total content, respectively. Despite the lower amount of terpenoids present (1.29%), experimental evidence shows an association between their presence and *Alternaria alternata* infection resistance in goji berry fruit [51]. It has been proposed that triterpenoids of cuticular waxes have a potential role in maintaining fruit integrity and postharvest quality and extending shelf-life because they provide mechanical toughness and protection against pathogen infections in lingonberry fruit (*V. vitis-idaea*) [64]. In agreement, the cuticular wax fraction of pear fruits, mainly composed of triterpenoids, inhibits *A. alternata* germination and growth in vitro, indicating that these compounds might contribute to antifungal protection against fungal pathogens in pear fruit [34]. 

It has been suggested that the increase in wax accumulation in the early stages of nectarine development could play a role in the resistance to fungus infection and water loss at harvest and during postharvest conditions, apparently due to the presence of triterpenoids. Oleanolic and ursolic acids appear to contribute to nectarines’ fruit resistance to brown rot caused by *Monilinia laxa* in the middle stages of development, but this resistance is not observed at the maturity stage. Authors argue that the lack of resistance showed at the mature stage could be due to the presence of microcracks in the fruit epidermis, which affect cuticle integrity, and a higher level of alkanes in the cuticle, which could serve as a carbon source favoring the growth of fungus [32]. 

The analysis of epicuticular wax components of mandarin “Satsuma” fruit (*C. unshiu*) showed that they could promote mycelial growth of *Penicillium digitatum*, whereas cutin components could inhibit conidial germination at different storage periods [25]. Furthermore, the tomato mutant DFD exhibits resistance to microbial pathogens even when the cuticular wax has been removed, which suggests a possible role of the cutin structure in the resistance to microbial pathogens in tomato fruit [12]. The studies mentioned above show that triterpenoids and cutin components, rather than VLC aliphatics, appear to have a more relevant function during the fruit defense to pathogen infection.

The biochemical metabolism of fleshy fruit during development and postharvest has been widely studied. Although several studies showed the pivotal role of cuticle biosynthesis in fleshy fruit quality [9,10], studies using genes and proteins to enhance fruit quality through cuticle modification are still scarce. These studies have allowed the identification of genes and proteins that can be modified to extend fleshy fruits’ shelf life [3,75]. Furthermore, this knowledge can contribute to generating technologies to improve fruit quality by increasing postharvest shelf-life, enhancing pathogen resistance, reducing physical disorders and reducing softening and water loss rates. Below, studies regarding the molecular pathway of cuticle biosynthesis in fleshy fruits and its relationship with softening, water loss and fruit quality are described.

## 4. The Molecular Pathway of Cuticle Biosynthesis

Cuticle composition is the phenotypic result of the activation of a specific biochemical pathway, which includes the participation of specific transcription factors, enzymes, transporters and RNA regulators in response to a *stimulus* [8,16]. Due to the considerable importance of cuticles in plant physiology, development and defense against biotic and abiotic stress, efforts to elucidate the molecular pathway of cuticle biosynthesis have been carried out, mainly in model plants such as Arabidopsis [13] and relevant crop plants such as wheat (*Triticum astivum*), rice (*Oryza sativa*), barley (*Hordeum vulgare*) and maize (*Zea mays*) [15]. 

In the case of fruits, most studies have been carried out in the model plant tomato (*S. lycopersicum*) [8,14,76]. Furthermore, by using massive RNA sequencing (RNAseq) and bioinformatic analysis, it has been possible to identify orthologous genes and proteins playing a role in cuticle biosynthesis in other fruits. However, some aspects of their regulation, the transport of their monomers across the cell wall and their macromolecular structure have not been clarified yet [5,16,77]. 

Both cutin and cuticular wax biosynthesis begin with C_16_ and C_18_ fatty acid (FA) precursors. Cutin biosynthesis involves the (i) synthesis of FA, (ii) FA oxidation, (iii) the export of the cutin monomers and (iv) their assembly [5]. Cuticular wax biosynthesis involves (i) FA elongation, (ii) functional groups modification (including primary alcohols and alkanes pathways) and iii) transport [11]. The mevalonic acid pathway synthesizes triterpenoid and sterol waxes [78]. In the next section of the review, genes and proteins involved in each of these processes are described, emphasizing discoveries carried out in fruits. 

### 4.1. Cutin Biosynthesis

The precursors C_16_ and C_18_ FA are synthesized in plastids, where they are bonded to an acyl carrier protein (ACP) forming a fatty acyl-ACP complex. In plastids, the enzyme ketoacyl ACP-synthase (KASIII) can synthesize C_18_ FA from C_16_ FA through the condensation of a malonyl-coenzyme A (malonyl-CoA) molecule. Fatty acyl-ACP thioesterase A (FATA) and B (FATB) release C_16_ and C_18_ FA from ACP [79]. After, they are exported to the cytoplasm, where long-chain acyl-CoA synthetase (LACS) esterifies the FA to a coenzyme A (CoA) molecule [80,81]. Fatty acyl-CoA (C_16_ and C_18_) are then transported to the endoplasmic reticulum (ER) where FA oxidation occurs [5]. 

Protein members of the cytochrome 450 family such as CYP86A carry out the terminal (ω) oxidation of FA [82]. Then, CYP77A carries out the mid-chain oxidation [83]. It has been demonstrated that CYP77A19 and CYP77A20 from *Solanum tuberosum* oxidize fatty acids in vitro. Besides this, it was proved that *CYP77A19* and *CYP77A20* expression partially restored the wild phenotype in an Arabidopsis cutin mutant [83]. Altogether, these reactions can lead to the synthesis of saturated and unsaturated ω-hydroxy FA, ω-midchain-dihydroxy FA, ω-9,10-trihydroxy FA and 9,10-epoxy ω-hydroxy FA [84]. In addition, the oxidation of the ω-hydroxyl group could be carried out, leading to the synthesis of α-ω-dicarboxylic FA. It has been suggested that a CYP86B carries out this reaction, but this has not been confirmed yet [5]. In vitro analysis has demonstrated that SlCYP86A69 plays a role in the ω-hydroxylation of octadecenoic acid in tomato fruits [85]. The mutation of the gen *SlCYP86A69* causes a deficient cutin and altered fruit surface structure [86]. 

Glycerol-3-phosphate acyltransferase (GPAT) transfers the acyl group from acyl-CoA to a glycerol-3-phosphate molecule, leading to the synthesis of 2-monoacylglycerol (MAG) cutin monomers [87,88]. The heterologous expression in yeast of *EgGPAT1* from *Echium pitardii*, homologous to AtGPAT4/AtGPAT8 from Arabidopsis, shows acyltransferase activity. Furthermore, the ectopic expression of *EgGPAT1* leads to an increase in the synthesis of cutin monomers in leaves of tobacco [87]. Besides this, it has been reported that *PpGPAT2* and *PpGPAT4* from *Physcomitrella patens* play a role in cuticle biosynthesis. A mutation in the gen *Slgpat6-a* abolishes the enzymatic activity of GPAT6 and leads to the alteration of the cuticle thickness, composition and properties of tomato, which suggests that this protein plays a central function in fruit cutin biosynthesis [89]. 

It has been hypothesized that MAGs are exported from the endoplasmic reticulum (ER) across the plasma membrane and cell wall through the activity of ATP Binding Cassette subfamily G (ABCG/WBC) and Glycosylphosphatidylinositol-Anchored Lipid Transfer Protein (LTPG) transporters, respectively [11]. It has been demonstrated that the ATP Binding Cassette transporters (ABCG), AtABCG32 from Arabidopsis and SlABCG42 from tomato can export 10,16-dihydroxy hexadecanoyl-2-glycerol, and the free fatty acids can export omega-hydroxy hexadecanoic acid and hexadecanedioic acid in vivo [90]. LTPG1 is required for the export of lipids in Arabidopsis, and it has been reported that LTPG2 is targeted to the plasma membrane, suggesting that it has an overlapping role with LTPG1 in cuticular wax deposition [91,92]. Nevertheless, the mechanism through which cutin monomers move across the cell wall is still unclear [77].

Once in the surface of epidermal cells, proteins belonging to the GDSL-motif lipase/esterase (GDSL) superfamily and cutin synthases (CUS) incorporate MAG into the cuticle matrix through esterification [5]. In tomato fruit, it was demonstrated that GDSL1 plays a role in the extracellular accumulation of cutin polyester. By immunolabeling experiments, it was shown that GDSL1 is trapped in the cuticle of tomato fruit. Further, the silencing of *GDSL1* leads to a reduction in ester bond cross-links in cutin [93]. Cutin monomers are esterified to form self-assembled particles named cutinsomes. CUS1 is an acyltransferase associated with the outer epidermal wall, which bonds cutin monomers contributing to cutin deposition in tomato fruit surface [94,95]. 

It has been hypothesized that cutin monomers are linked to the hydroxyl groups of the cell wall components by covalent ester and/or ether bonds [5]. However, the sites of deposition of the cutin polyesters on the outer faces of the cell wall are still unknown [77]. In Figure 1, the chemical structure of the main cutin components found in fleshy fruit cuticles and the main enzymes that could carry out the synthesis of these components are shown.

### 4.2. Cuticular Wax Biosynthesis

#### 4.2.1. Fatty Acids Elongation

Cuticular wax biosynthesis is carried out in the ER and includes FA elongation, functional group modification and wax transport. Like cutin monomers, wax biosynthesis uses the precursors FA-CoA (C_16_ and C_18_) and malonyl-CoA, generated by LACS, and acetyl-CoA carboxylase (ACCase), respectively [11]. A process of elongation is carried out in the ER membrane by proteins belonging to the fatty acid elongase (FAE) multienzyme complex. Each elongation cycle involves four consecutive reactions that perform the addition of two-carbon atoms to the acyl chain [11]. These reactions consist of (i) the condensation of malonyl-CoA to the FA-CoA molecule by the enzymes beta-ketoacyl-CoA synthases (KCS) [97], (ii) reduction of the beta-ketoacyl-CoA by beta-ketoacyl-CoA reductase (KCR), synthesis of enoyl-CoA by beta-hydroxyacyl-CoA dehydratase (HCD) and reduction of enoyl-CoA by enoyl-CoA reductase (ECR) [98]. Thereby, VLCFAs with 20 or more carbon atoms are synthesized from FA-CoA C_16_ and C_18_ monomers [99]. 

The Arabidopsis genome includes 21 genes predictively encoding KCS, from which ECERIFERUM6 (CER6/KCS6) carries out the most important role in wax biosynthesis [8]. Furthermore, it was demonstrated that CER6 (KCS6) is essential for the synthesis of VLCFA longer than C_28_ in tomato and appears to be involved in the generation of branched VLCFA [100]. It was shown that MdKCS2 from apple is located at the ER and its gene has a higher expression in apple pericarp. Besides, it was proved through ectopic expression that MdKCS2 increases wax content in Arabidopsis [101].

Through the analysis of *Citrinae* species, including the cultivated citrus *C. clementina* and *C. sinensis*, 96 genes encoding KCS have been identified, from which CsKCS2 and CsKCS11 proteins are located in the ER, and their genes increase their expression in fruits at the ripening stage, suggesting that they are involved in the accumulation of fruit cuticular wax during the ripening of citrus [102]. Once the FAs have been elongated, both VLC alcohols and VLC aldehydes can be generated through the reduction of VLCFA by fatty acyl-CoA reductases (FAR) enzymes, through the primary alcohols and alkanes pathways, respectively [8].

#### 4.2.2. Primary Alcohols Pathway

The acyl-reduction pathway, also known as the primary alcohols pathway, carries out the reduction of the carboxylic group from VLCFA to form primary alcohols [8]. In agreement with that previously reported for plants [11], wax composition analysis in fleshy fruits shows that this pathway predominantly synthesizes primary alcohols in even-numbers carbon atom wax compounds. This reaction is mainly catalyzed by the alcohol-forming fatty acyl-CoA reductase named CER4 [103]. The expression of *CsCER4*, homologous to *AtCER4* from Arabidopsis, changes during cucumber fruit development. The different expression level of *CsCER4* between glossy type and waxy type cucumber suggests its role in wax biosynthesis in this fruit [104]. 

In the primary alcohols pathway, primary alcohols are used to synthesize esters through esterification to FA by wax synthetases (WS) enzymes. Wax synthase/diacylglycerol acyltransferase 1 (WSD1) is a WS located at the ER that catalyzes the synthesis of wax esters using mainly C_16_ acyl-CoA as precursors [105]. Further, it was suggested that CER17 can perform the desaturation of VLC acyl-CoA leading to the synthesis of unsaturated primary alcohols [106]. Thereby, the primary alcohols pathway leads to the generation of saturated and unsaturated VLC primary alcohols and wax esters [11].

#### 4.2.3. Alkanes Pathway

The alkanes pathway uses as precursors the aldehydes generated by the reduction of the carboxylic group from VLCFA. In accordance with that previously reported for plants [11], wax composition analysis in fleshy fruits shows that this pathway predominantly synthesizes alkanes in odd-numbers carbon atom wax compounds. A decarbonylation of aldehydes is carried out to form alkanes by the enzymatic complex CER1-CER3/WAX2, leading to the elimination of one carbon atom in the carbon chain [107]. It has been proved that the gene *CsWAX2* located at the ER from cucumber (*Cucumis sativus* L.), which is homologous to *AtWAX2* from Arabidopsis, is highly expressed in the epidermis and is involved in wax biosynthesis and plant response to abiotic and biotic stress [108]. 

Cucumber fruits with a waxy phenotype have a higher expression of the gene *CsCER1*. The CsCER1 protein is located at the ER, and its gene is specifically expressed in the fruit epidermis. Drought, low temperature and abscisic acid induce the expression of *CsCER1*. Furthermore, abnormal expressions of *CsCER1* induce alterations of cuticle permeance, drought resistance and VLC alkanes biosynthesis, suggesting that it can carry out a pivotal role in VLC alkanes synthesis, especially in response to abiotic stress [109]. In the alkanes pathway, the midchain oxidation of alkanes can be carried out by the cytochrome P450 enzyme (CYP96A15) mid-chain alkane hydroxylase (MAH1) to generate secondary alcohols and subsequently ketones [110]. Thereby, alkanes, aldehydes, secondary alcohols and ketones can be generated by this pathway [11]. 

#### 4.2.4. Triterpenoid Wax Biosynthesis

The mevalonic acid pathway synthesizes triterpenoid and sterol waxes. The cyclization of 2,3-oxidosqualene leads to the synthesis of triterpenoids by the activity of oxidosqualene cyclase (OSC) enzymes [78]. Beta-amyrin synthase (BAS) activity leads to the synthesis of lupeol, alpha and beta-amyrin [111]—one of the most prevalent terpenoid compounds in fleshy fruit cuticles. In tomato fruits, triterpene synthase 1 (SlTTS1) and 2 (SlTT2) catalyze the synthesis of alpha and beta-amyrins [112]. Then, a carboxylic functional group is introduced to beta and alpha-amyrin by proteins of the cytochrome 450 family CYP716A, leading to the synthesis of oleanolic and ursolic acid, respectively [78,111]. Regarding cuticle biosynthesis, the introduction of carboxylic groups could pave the way for further polymerization by enzymes such as acyltransferases. It has been shown that CYP716A46 and CYP716A44 from tomato can synthesize oleanolic and ursolic acid by the C-28 oxidation of beta and alpha-amyrins, respectively [111]. 

### 4.3. Transport of Cuticle Components

It has been hypothesized that waxes are transported from the ER to the plasma membrane (PM) through Golgi vesicles, then exported to the apoplast by the activity of ABCG and LTPG transporters [16]. Like cutin precursors, it has been suggested that the transporters from the ABCG subfamily AtABCG11/CER5, AtABCG12, AtABCG13 and AtABCG32 are required for wax export in Arabidopsis [8]. It has been suggested that cuticular components are secreted through vesicles derived from the trans-Golgi networks that fuse with the PM. Further, other hypotheses suggest that waxes can be transported directly from the ER through ER-PM contact sites [11]. 

More recently, a passive mechanism of cuticular component transport involving a phase-separation process have been proposed [77]. This hypothesis suggests that the highly hydrophobic waxes could cross the cell wall through association with an aggregate of amphiphilic cutin monomers [77]. Nevertheless, the mechanism of transport and the molecular organization between cutin, waxes and the cell wall components have not been elucidated yet. Omics sciences facilitate the identification of transcripts and proteins involved in relevant biochemical pathways in fruits. Recently, efforts to identify genes and proteins involved in cuticle biosynthesis and transport in fleshy fruits have been carried out. 

In Figure 2, the structure of the main cuticular wax components found in fleshy fruit cuticles and enzymes that could carry out their synthesis are shown. 

In the next section of the review, a transcriptomic and proteomic analysis complemented with differential gene expression analysis such as qRT-PCR, contributing to the elucidation of the molecular pathway of cuticle biosynthesis, regulation, transport and its association with fruit quality, is described.

## 5. An Overview of the Current Status in the Elucidation of Molecular Mechanism of Cuticle Biosynthesis in Fleshy Fruits, Its Regulation, and Physiological Function

Through laser microdissection (LMD), transcriptomic analysis and differential expression gene protocols, it has been possible to identify genes that express in peel and epidermis, playing a role in cuticle biosynthesis and stress response in fruits [113]. An increase in the expression of *SlCER6*, *GDSL*, *LTP* and MD-2-related lipid recognition domain-containing (ML) protein (*MD2*) correlates with the stages of major cuticle deposition during tomato fruit development [114]. Furthermore, by pyrosequencing and LMD, epidermal-specific transcripts related to cuticle biosynthesis in tomato fruit epidermis were identified, including *CYP450*, *AtCER10*, *LACS*, *KCS6*, *CER2*, *CER6*, *CER1* and *LTP*, and the transcriptional factors SHINE1/WAX INDUCER 1 (*SHN1/WIN1)*, zinc finger *(ZNF)*, cutin deficient 2 *(CD2)* and *MYB* [76]. 

Thirteen genes specific to the exocarp and correlating with the cuticle deposition pattern of sweet cherry fruits (*P. avium*) were identified, including *PaLipase, PaLTPG1,* the CYP450 aberrant induction of type three genes 1 *(PaATT1)*, LACERATA *(PaLCR)*, *PaGPAT4/8*, *PaLACS2*, *PaLACS1* and *PaCER1* and the transcriptional factors *PaWINA* and *PaWINB* [115]. The major deposition of cuticle in sweet cherry fruits and the major expression of the genes *PaLIPASE*, *PaFATB*, *PaLACS9*, *PaLTPG1*, *PaWBC11*, *PaWINA* and *PaWINB* occurs at the earlier stages of development, strongly suggesting their participation in cuticle biosynthesis in cherry fruits. Nevertheless, the expression of *PaCER5*, *PaLACS9*, *PaLTPG1* and *PaWBC11* increases at later stages, despite the lack of a significant increase in cuticle deposition [116]. 

Through the transcriptomic analysis of mango peel during fruit development, transcripts orthologous to the transcriptional factors MiSHN1 and MiCD2, the enzymes MiCER1, MiCER2, MiCER3, MiKCS2, MiKCS6, MiCUS1 and MiCUS2 and the transporters MiWBC11, MiLTP1, MiLTP2, MiLTP3 and MiLTPG1 related to cuticle biosynthesis were identified. The higher expression of transcripts related to cuticle biosynthesis was shown at 153 DAF in ripening stages, and it was shown to correlate with the major cuticle deposition through mango development [117].

Through metabolomic and transcriptomic analysis, 27 genes related to cuticular wax biosynthesis and regulation from “Yuluxiang” pear fruit were identified, including *PbrLACS1*, *PbrMAH1*, *PbrLTP3*, *PbrDGAT1*, *PbrWIN1*, *PbrKCS2*, *PbrKCS4*, *PbrKCS6*, *PbrKCS10*, *PbrECR*, *PbrCER9* and *PbrKCR1*. Furthermore, 12 genes coding for transcriptional factors and a new gene tentatively coding for a beta-amyrin synthase were identified. Deposition patterns during pear development positively correlate with the expression of these cuticle-related genes. These findings allowed the identification of the cuticle biosynthesis pathway in pear fruit [118]. 

At the green expanding stage in mandarin epidermis (*C. clementina)*, the most expressed genes are related to cuticle biosynthesis, flavonoid biosynthesis and defense response [119]. During “Newhall” navel orange development, the expression of genes involved in cutin and wax biosynthesis significantly increases in later development stages. Out of these, a transcription factor MYB (GL1-Like) appears to regulate the wax synthesis by controlling the expression of *CER, KCS,* and *LACS* genes [119]. The analysis of “Navelate” sweet orange fruits and its abscisic acid (ABA) biosynthesis-impaired mutant “Pinalate” suggest that ABA could regulate wax biosynthesis genes. The expression levels of the genes analyzed in this study were consistent with the amounts of VLC aliphatics and terpenoids, which enhance the cuticle metabolism pathway for sweet orange [22]. 

It was shown through the transcriptomic and metabolomic analysis of “Newhall” navel orange and its glossy mutant “Gannan No. 1′ that the down-regulation of the genes *FATB, LACS1, LACS2, KCS, FAR2* and *CER3* and the transcription factor coding gene *MYB16* was associated with a decrease in VLC aliphatic [120]. RNAseq analyses of “Newhall” navel orange peel and its glossy mutant “Ganqi 3’ suggest that the glossy phenotype could be due to the decrease in the expression of *CsACC1, CsCAC3, CsKASI*, four *CsLACS*, six *CsKCS*, three alkane-forming pathway genes (*CsCER1-LIKE* and *CsCER3*), five alcohol-forming pathway genes (*CsCER4-LIKE* and *CsFAR2-LIKE*) and fourteen ABCG transporters genes, including *CsABCG11-LIKE, CsABCG15,* and *CsABCG32* [121]. 

Analysis by qRT-PCR showed that genes encoding cuticle biosynthesis enzymes and the transcriptional factors encoding genes *CsMYB16*, *CsMYB94* and *CsMYB96* have a low expression at 60 days after flowering (DAF), increase at 150 DAF and then decrease at 210 DAF in both “Newhall” navel orange and “Ganqi 3”. In contrast, the expression of *CsLACS9*, *CsKCS2-LIKE1*, *CsKCS2-LIKE2*, *CsFAR2-LIKE2* and *CsCER7* increased continuously during development. Nevertheless, at 150 and 210 DAF, “Ganqi 3’ exhibited a low expression of almost all cuticle-related genes analyzed [122]. Like “Ganqi 3’, the navel orange glossy mutant “glossy Newhall” exhibits a decrease in the expression of genes related to wax biosynthesis and export. Besides, a loss of epicuticular wax crystals is shown in “glossy Newhall” [21]. Authors suggest that the glossy phenotype in “Ganqi 3’ and “glossy Newhall” is due to a reduction in the VLC aliphatic compounds due to the decrease in the expression of wax-related genes observed [21,121].

Transcripts putatively involved in cuticle biosynthesis and with tissue-specific expression were identified in apple fruit “Florina” and “Prima”, including *LACS2*, *KCS7/2*, FIDDLEHEAD (*FDH*), HCD/PASTICCINO2 (*PAS2*), *CER10*, *CER1*, *CER4*, *LCR*, *WBC11*, *LTPG1* and *WIN1*. A higher expression in the peel than in pulp was confirmed, and differences were observed between “Florina” and “Prima” cultivars. With these data, it was suggested that they can be associated with the known difference in wax composition between “Florina” and “Prima” cultivars [122]. The metabolomic and transcriptomic profiling of habanero peppers (*Capsicum chinense* Jacq.) genotypes PI 224448 and PI 257145, which exhibit low and high levels of cutin amounts, respectively, show that GDSL lipase, GPAT6, CYP86A, CYP77A and the transcription factors SHN1, ANTHOCYANINLESS2 (ANL2) and homeodomain GLABROUS 1 (HDG1) could be contributing to this phenotype [65]. 

Seventy-nine genes potentially related to wax biosynthesis were identified through transcriptomic analysis of blueberries. Differential expression analyses between waxy and non-waxy blueberries led to the identification of a FATB gene tentatively associated with the waxy coating phenotype [123]. Analysis of the glossy type of bilberry suggests that the contents of fatty acids and ketones in cuticle composition affect the amounts of crystals and the glaucous phenotype in this fruit. Furthermore, the specific expression of the genes *CER26-like*, *FAR2*, *CER3-like*, *LTP*, *MIXTA* and *BAS* suggest their role in wax biosynthesis and the glaucous phenotype’s appearance in the bilberry skin [54]. 

Transcripts coding for the transcriptional factors MYB42, MYB52 and MYB93, the enzymes LACS1, CYP86A, GPAT6, KCS4, KCS2, KCS10, CER1, CER3, CER6 and FAR and the transporters ABC, WSD and WBC11, related to cuticle biosynthesis, have been identified in the epidermis of apple fruit cultivar “Cox Orange Pippin”. A decrease in cuticular waxes, ursolic and oleanolic acid, along with low expression of *LACS1/CER8, LCR, GPAT6, KCS2, 6* and *10* is shown on the russeted patches of the semi-russeted apple variety “Cox Orange Pippin” [124], suggesting a relationship between the expression of genes related to wax biosynthesis and the russeting development in fruits.

Consistent with the changes shown in cuticle composition, genes involved in the synthesis of C_16_ and C_18_ FA and *CER1* are induced by oleocellosis in lemon fruit, suggesting that the increase in the deposition of fatty acids and the content of VLC alkanes in cuticles is associated with the appearance of oleocellosis [26]. 

“Kordia” sweet cherry cultivar, which shows a cracking resistance phenotype, has a higher expression of *PaWINB*, *WS* and *PaKCS6* during the fruit setting stage than the cracking susceptible “Bing” cultivar. Authors suggest that the cuticle deposition function of these genes could be involved in the tolerance to cracking observed in the “Kordia” cultivar [125].

Cold storage at 0 °C decreases the expression of the genes *PpCER1*, *PpLACS1* and *PpLipase* in “October Sun” peach fruit, but after five days of room temperature storage, the expression levels of *PpLACS1* and *PpLipase* increase. In addition, CO_2_ and heat treatment restored the expression of *PpLACS1* of cold-stored fruits to similar levels to those of harvest conditions [74], suggesting a regulation of cuticle metabolism in response to CO_2_, heat and cold storage conditions. At postharvest, cold storage and 1-MCP treatment decreased the cuticular wax density, delayed wax crystal melting and senescence and reduced the expression of the genes *MdCER6*, *MdCER4* and *MdWSD1* in apple “Starkrimson” fruits, whereas ethephon induced the opposite effect. Authors suggest an effect of ethylene in cuticular wax composition and crystal morphology through the regulation of *MdCER6* and the alcohol forming pathway in apple fruit during postharvest cold storage [126]. 

During tomato fruit development, the genes related to cuticle biosynthesis *CUS1, GPAT4, CER6,* triterpene synthase 2 (*TTS2*) and cutin deficient 3 (*CD3*) were expressed at higher levels during mature green than during red ripe stages. At standard conditions, AC tomato fruits have a lower expression of these genes than DFD; nevertheless, water stress treatment significantly increases their expression in AC, but not in DFD [9]. Water deficit increases the total wax and esters amount and the expression of the genes *WSD1, CER1, CER2, CER3, CER4* and *CER10* in grape berry fruits. Additionally, an increase in the wax compound with a chain length from C_40_ to C_50_ is shown. However, despite the change in cuticle composition observed, no significant difference in fruit transpiration was shown [44]. 

The expression of *MIXTA*, which codes for transcription factor type MYB, occurs predominantly in epidermal cells of tomato fruit. It has been shown that *MIXTA* silencing leads to a thinner cuticle, an increased susceptibility to pathogens and water loss, accompanied by a decrease in the expression of the genes *CYP77A, CYP86A, LACS2, GPAT4, ABCG11, ABCG32, GDSL, KCS3* and *SHN3*, suggesting that MIXTA regulates wax biosynthesis and could modulate the integrity of tomato fruit during postharvest [127]. It was reported that the gene MdKCS2 shows a large expression in apple pericarp. Besides this, it was proved through ectopic expression that MdKCS2 improved drought resistance in Arabidopsis by changing epidermal permeability and increasing wax deposition. In agreement with these findings, it was shown that drought and salt stress induce the expression of MdKCS2 [101]. 

In pepper fruit, two linked quantitative trait loci (QTL) associated with reduced post-harvest water loss traits have been identified. Transcriptomic analysis showed that a higher expression of the genes *CER1, CER3, LTP, FAR* and *CYP96A/MAH1*, along with a decrease in FA and an increase in the amount of iso-alkanes in the cuticle, could be related to post-harvest water loss in pepper. Besides, a relationship between post-harvest water loss, a delayed over-ripening on the plant and reduced fruit softening after storage is suggested [128]. Efforts to elucidate the molecular pathway of fleshy fruits and the regulation and molecular response during fruit developmental stages in response to environmental conditions and treatments on plant or during postharvest conditions will contribute to the design of biotechnological strategies to develop plants adapted to grow and produce fleshy fruits under conditions of limited water supply, without affecting fruit productivity and quality.

## 6. Concluding Remarks

The cuticle composition, amount and biosynthesis pattern during fruit development are clearly different among various fruits, suggesting a role of fruit adaptation to the different environments in which the different fruit species evolved. This composition has been correlated with the physiology of fruit phenotypes in different studies. The data generated by the analysis of genes playing a role in cuticle biosynthesis further support the involvement of cuticle-specific components with a given fruit phenotype. This also has allowed us to move towards the elucidation of the molecular mechanism of cuticle biosynthesis. The study of the genes involved in cuticle biosynthesis in different fruits seems to suggest that there is a universal molecular mechanism of fruit cuticle biosynthesis that is active in the different fruit species.

The physiology of fruit cuticles is broad and includes responses to pathogen attack, biotic and abiotic stress, fruit quality and a role in development of physiological disorders. With all these data available along with those that are currently being generated, in the future, it will be possible to design fruits with stronger resistance to pathogen attack, that are less prone to develop physiological disorders, that have a low rate of weight loss and softening and that have a longer shelf life. This in turn will boost the sales of fruits to international markets, which is an important economic activity for developing countries such as Mexico.

## Figures and Tables

**Figure 1 plants-11-01133-f001:**
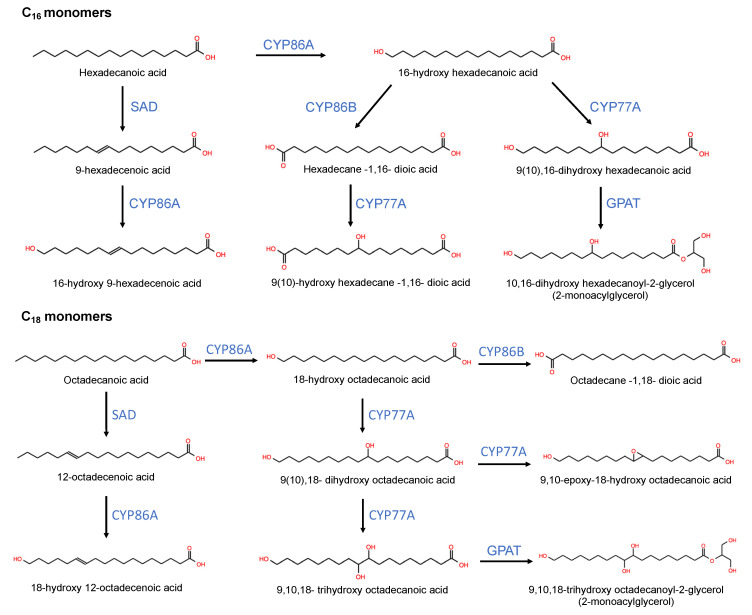
Biosynthesis pathway and chemical structure of common cutin monomers found in the fleshy fruit cuticle. Only the gene subfamily’s main name is included for reactions involving multiple paralogs. 9-hydroxy fatty acid structures are used to represent both 9 and 10 hydroxy fatty acid monomers. The main C_16_ and C_18_ monomers are used to exemplify the structure of 2-monoacylglycerol monomers. Abbreviations: Cytochrome P450 subfamily 86A, 86B and 77A (CYP86A, CYP86B and CYP77A, respectively); StearoyI-ACP desaturase (SAD); Glycerol-3-phosphate acyltransferase (GPAT). The figure was built based on the literature [5,8,16,82,83,84,85,86,87,88,89]. Chemical structures were drawn with the JSME Molecular Editor [96].

**Figure 2 plants-11-01133-f002:**
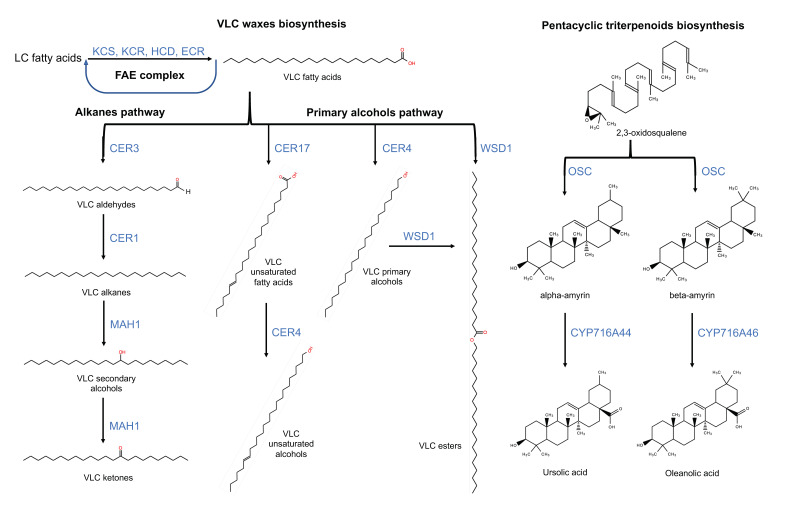
The biosynthesis pathway and chemical structure of common cuticular wax compounds found in the cuticle of fleshy fruits. Only the gene subfamily’s main name is included for reactions involving multiple paralogs. Twenty-four chain length carbon atoms are used to exemplify the main very long chain (VLC) wax compounds synthesized by the primary alcohols pathway. Twenty-three chain length carbon atoms are used to exemplify the main VLC wax compounds synthesized by the alkane pathway. Abbreviations: long-chain (LC); fatty acid elongase multienzyme complex (FAE); beta-ketoacyl-CoA synthase (KCS); beta-ketoacyl-CoA reductase (KCR); beta-hydroxyacyl-CoA dehydratase (HCD); Enoyl-CoA reductase (ECR); Fatty acyl-CoA reductase/ECERIFERUM1, 3, 4 and 17 (CER1, CER3, CER4 and CER17, respectively); Wax synthase/diacylglycerol acyltransferase 1 (WSD1); Mid-chain alkane hydroxylase (MAH1); Oxidosqualene cyclase (OSC); Cytochrome P450 subfamily 716A44 and 716A46 (CYP716A44 and CYP716A46, respectively). The figure was built based on the literature [6,8,11,13,16,97,98,99,100,101,102,103,104,105,106,107,108,109,110,111]. Chemical structures were drawn with the JSME Molecular Editor [96].

**Table 1 plants-11-01133-t001:** The main cuticle components present in fleshy fruits of different species.

Fruit	Layer	Main Components	References
**Tomato** (*Solanum lycopersicum*)	Cuticular wax	Alkanes (C_29_ and C_31_) and triterpenoids (amyrins)	[18,19,20]
	Epicuticular	VLC aliphatic compounds	
	Intracuticular	Pentacyclic triterpenoids	
	Cutin	9(10),16-dihydroxy hexadecanoic acid	
**Orange** (*Citrus sinensis*)	Cuticular wax	Fatty acids (C_26_), alkanes (C_31_), primary alcohols, aldehydes and triterpenoids	[21,22,23]
	Epicuticular	Fatty acids and alkanes	
	Intracuticular	Fatty acids, triterpenoids and primary alcohols	
	Cutin	*cis*-9-hexadecenoic acid and cinnamic acid	
**Mandarin** (*Citrus unshiu*)	Cuticular wax	Aldehydes, alkanes, fatty acids and primary alcohols	[24,25]
	Epicuticular	Fatty acids, followed by alkanes and terpenoids	
	Intracuticular	Terpenoids, followed by alkanes and fatty acids	
	Cutin	Cinnamic acids, hexadecanedioic acid (C_16_) and hexadecanoic acid (C_16_)	
**Lemon** (*Citrus limon*)	Epicuticular	Alkanes (C_31_), aldehydes, alcohols and fatty acids	[26]
**Apple** (*Malus domestica*)	Cuticular wax	Fatty acids, alkanes (C_29_), triterpenoids (ursolic acid) and primary alcohols	[27,28,29]
	Cutin	9(10),16-dihydroxy hexadecenoic acid	
**Sweet cherry** (*Prunus avium*)	Cuticular wax	Triterpenes (ursolic acid), alkanes (C_29_) and alcohols	[30]
	Cutin	9(10),16-dihydroxy-hexadecanoic acid Monocarboxylic, dicarboxylic and ω-hydroxylated octadecanoic acids.	
**Nectarine/Peach **(*Prunus persica*)	Cuticular wax	Triterpenoids (oleanolic and ursolic acid), alkanes (C_23_ and C_25_) and fatty acids	[31,32]
	Cutin	Mono-carboxylic, α,ω-dicarboxylic and ω-hydroxylated fatty acids 18-hydroxyoleic acid	
**Drupe fruit** (*Prunus laurocerasus*)	Cuticular wax	Triterpenoids (ursolic acid), fatty acids, alkanes and primary alcohols	[33]
	Cutin	9(10),-dihydroxy hexadecanoic acid, 9,10-epoxy 18-hydroxy octadecanoic acid and 9,10,18-trihydroxy octadecanoic acid	
**Pear** (*Pyrus* spp.)	Cuticular wax	Alkanes (C_29_), primary alcohol (C_30_), terpenoids and fatty acids	[34,35,36,37]
**Berries** (*Vaccinium* spp.)	Cuticular wax	Triterpenoids (oleanolic and ursolic acid), beta-diketones and fatty acids	[38,39,40,41]
**Grape** (*Vitis vinifera*)	Cuticular wax	Triterpenoid (oleanolic acid), primary alcohols, fatty acids and esters.	[42,43,44,45]
	Epicuticular	Primary alcohols, fatty acid, esters and terpenoids	
	Intracuticular	Triterpenoid (oleanolic acid)	
**Pepper** (*Capsicum* spp.)	Cuticular wax	Alkanes (C_29_ and C_30_), triterpenoids (amyrins), phytosterols, fatty acids and primary alcohols	[46]
	Cutin	9(10),16- dihydroxy hexadecanoic acid	
**Olive** (*Olea europaea*)	Cuticular wax	Triterpenoids (oleanolic acid), primary alcohols (C_26_) and fatty acids (C_26_)	[47]
	Cutin	9(10),16-dihydroxy hexadecanoic, 9,10,18-trihydroxy octadecenoic and 9,10,18-trihydroxy octadecanoic acids	
**Guava** (*Psidium guajava*)	Cuticular wax	Fatty acids (C_28_), primary alcohols (C_30_) and terpenoids (uvaol, ursolic acid and maslinic acid)	[48]
	Cutin	9(10),16-dihydroxy hexadecanoic acid and 9,10-epoxy-18–hydroxy octadecanoic acid	
**Pitahaya** (*Hylocereus polyrhizus*)	Cuticular wax	Triterpenoids, alkanes (C_31_ and C_33_) and fatty acids	[49]
	Cutin	9(10),16-dihydroxy hexadecanoic acid and 9,10-epoxy-18-hydroxy octadecanoic acid	
**Jujube** (*Ziziphus jujuba*)	Cuticular wax	Fatty acids, primary alcohols and alkanes	[50]
**Goji berry** (*Lycium barbarum*)	Cuticular wax	Fatty acids, alkanes and primary alcohols	[51]

**Table 2 plants-11-01133-t002:** Cuticle composition changes recorded during fleshy fruits development.

Fruit	Scientific Name	Observation	References
**Tomato**	*Solanum lycopersicum* L.	Continuous increase of alkanes, triterpenoids, and cutin.	[18]
**Orange**	*Citrus sinensis* L. Osbeck	Increasing alkanes amount. Decreasing fatty acids and aldehydes amount. Increasing epicuticular wax, terpenoid and hentriacontane amounts at the earlier stage.	[22]
**Apple**	*Malus domestica*	Increasing cuticular wax, nonacosane and heptacosane amounts.	[27]
**Sweet cherry**	*Prunus avium* L.	Decreasing of triterpenes and cutin amount.	[30]
**Nectarine**	*Prunus persica* L. Batsch	Increasing triterpenoids and cutin amounts at earlier stages, then they decrease until maturity. Increase in alkanes amount at the later stages.	[32]
**Drupe fruit**	*Prunus laurocerasus* L.	Increasing cutin, triterpenoids and cinnamic acid amounts at later stages.	[33]
**Pear**	*Pyrus bretschneideri*	Increasing fatty acids amount at earlier stages and then decreases. Continuous increase of triterpenoids amount.	[35]
**Blueberry**	*Vaccinium corymbosum* and *V. ashei*	Continuous increase of total wax and triterpenoids amounts at ripening. Decrease in the relative content of diketones. Increase in the relative content of aldehydes, primary alcohols, fatty acids and alkanes.	[40]
**Bilberry**	*Vaccinium myrtillus*	Decrease in triterpenes amount. Increase in aliphatic compounds amounts.	[54]
**Grape**	*Vitis vinifera*	Increasing triterpenoid, primary alcohols and aldehydes amounts at early stages and decreasing at ripening. Increase alkyl esters and fatty acids amount at later stages.	[42,43,44]
**Olive**	*Olea europaea*	Increasing in very long chain of acyclic, ω- hydroxy fatty acids and ω- mid-chain dihydroxy fatty acids amounts. Increasing in average chain length of the compounds. Decreasing of C_16_/C_18_ ratio of cutin monomers.	[47]
**Mango**	*Mangifera indica* L.	Increasing epicuticular wax and cutin amounts.	[55]

**Table 3 plants-11-01133-t003:** Cuticle composition changes in response to different postharvest storage conditions of fleshy fruits.

Fruit	Conditions	Observations	Ref.
	*Room temperature storage*		
** Peach ** (*P. persica* L. Batsch.)	5 days (20 °C).	Increasing of wax and cutin amount.	[31]
**Sweet orange** (*C. sinensis*)	40 days (25 °C).	Continuous increasing of epicuticular wax, triterpenoids and nonacosane amounts Increasing of cutin at 20 and 40 days. Decreasing of fatty acids amount.	[23]
**Mandarin** (*C. unshiu*)	40 days (25 °C)	Increasing epi- and intracuticular waxes amounts after 20 days, but decreasing after 40 days. Decreasing of terpenoids, fatty acids, and cutin amounts. Increasing of alkanes amount.	[25]
**Apple** (*M. domestica*)	49 days (25 °C)	Decreasing of wax, alkanes and primary alcohols amounts. Increasing in fatty acid proportion.	[59]
** Apple ** (*M. domestica*)	8 months in CA, DCA-CF, and DCA-RQ (20 °C)	Increasing wax concentration from 7 to 14 days. Increasing of unsaturated fatty acids, cis-11,14-eicosadienoic acid, nonacosane and tetracosanal amounts.	[71]
	*Cold storage*		
**Sweetcherry** (*P. avium* L.)	5 days (0 °C)	Increase in cuticle amount.	[61]
** Blueberry ** (*V. corymbosum* and *V. ashei*)	30 days (4 °C)	Decreasing of total wax content.	[40]
**Sweet orange** (*C. sinensis*)	40 days (4 °C).	Epicuticular wax amount increases at 30 days then decreases at 40 days. Cutin amount decreases continuously. Triterpenoids amount increase continuously at 20 days and then decrease at 40 days.	[23]
**Korla pear** (*Pyresbretschnei deli*)	90 days (0 °C)	Increasing of wax content during 30 days, but decreasing at day 90.	[68]
**Apple** (*M. domestica*)	140 days (0 °C)	Increasing total wax content from day 0 to day 80, then decreases at day 140.	[28]
** Apple ** (*M. domestica*)	7 months (0 °C)	Decreasing of total cuticular wax, nonacosane (C_29_) and heptacosane (C_27_) amounts. Increasing of nonacosan-10-ol, nonacosan-10-one and hexadecanoic acid amounts.	[27]
** Asian pear ** (*P. sinkiangensis* and *P. bretschneideri*)	7 months (3 °C)	Decreasing of total wax and variety of wax compounds. Decreasing primary alcohols amounts. Increasing alkanes amount.	[36]
	* Treatments with Ethylene and inhibitors of ethylene action *		
**Apple ** (*M. domestica*)	140 days with ethephon (0 °C)	Accelerating of total wax and VLC aliphatic deposition. Increasing octacosanoic acid.	[28]
**Apple ** (*M. domestica*)	140 days with 1-MCP (0 °C)	Delaying of total wax and VLC aliphatic deposition. Decreasing of octacosanoic acid.	[28]
**Apple** (*M. domestica*)	7 months (0 °C) with 1-MCP	Decreasing of nonacosan-10-ol, nonacosan-10-one and hexadecanoic acid.	[27]

**Abbreviations:** Controlled atmosphere (CA); Dynamic controlled atmosphere by Chlorophyll Fluorescence (DCA-CF); Dynamic controlled atmosphere by Respiratory Quotient (DCA-RQ); 1-methylcyclopropane (1-MCP).

**Table 4 plants-11-01133-t004:** Cuticle composition associated with physical skin disorders of fleshy fruits.

Fruit	Physical Phenotype	Observation	References
	*Glossy phenotype*		
**Orange “Newhall”** (*C. sinensis*)	Glossy mutant “Glossy Newhall”	Low amounts of aldehydes, alkanes, and wax crystals during fruit development. Less amount of epicuticular wax at later stages of development.	[21,56]
**Bilberry** (*V. myrtillus*)	Bilberry “Glossy Mutant”	A high proportion of triterpenes, and a lower proportion of fatty acids and ketones.	[54]
	*Physical disorder*		
**Pear “Dangshansuli”** (*P. bretschneideri*)	Russet mutant “Xiusu”	Low content of alkanes and high content of alcohols during development and ripening.	[35]
**Jujube “Popozao”** (*Z. jujuba Mill.*)	Cracking-susceptible “Hupingzao”	Low amount of total wax, alkanes, and aldehydes with a chain length greater than C_20_, and high amount of fatty acids.	[50]
**Lemon “Yunning number 1”** (*C. limon*)	Oleocellosis	High amount of alkanes (especially C_29_) and a low amount of aldehydes (especially C_32_)	[26]

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
