# Peer review of "Molecular Biology, Composition and Physiological Functions of Cuticle Lipids in Fleshy Fruits"

_plants, 2022, doi:10.3390/plants11091133_

Round 1

Reviewer 1 Report

This review addresses the important roles of cuticle of fruits in both preharvest and postharvest stages. This review is very comprehensive and covered all of the essential roles of cuticle in flesh fruits. It is a very useful article for researchers who work with extending shelf-life of fruits and the biosynthesis of cuticles. This topic is quite original and does add a lot to the other published materials. The review covered a lot of fruits and several vegetables. The conclusions are consistent with evidence and arguments. They certainly address the main question posed.

This review was well written. I made some suggested edits. Please refer to the attached file for more information.

Author Response

Dear Reviewer 1,

Thank you for your comments and suggestions. Regarding the suggestions in the manuscript edition, we have checked them, and did the changes following carefully the suggestions. In the Microsoft word file attached, you can see that all the suggestions have been edited and corrected. Thank you very much for your valuable observations.

Reviewer 2 Report

  1. In the manuscript, scientific names of plant species should be italic.
  2. When describing fleshy fruit cuticle compositions, different cultivars of certain fruit, like apple, present obvious differences in cuticle components. What are the similarities of cultivars sharing similar components? Do they share similar geographical location?
  3. Are the cuticle compositions related with the original flower organ where the fleshy fruit develop from? If it is related, it would be better to describe the original flower organs of each fleshy fruit.
  4. When describing the molecular pathway involved, please make it more clear in which species the molecular knowledge was obtained.

Author Response

Dear Reviewer 2,

Thank you very much for your comments and suggestions. Next, we are including a  reply to each of your comments and suggestions.

Regarding observation number 1:

  1. In the manuscript, scientific names of plant species should be italic.

In the original manuscript file, which is in the format .docx, all the scientific names of plant species are in italic font type,  but in the manuscript file in format .pdf, the scientific names of species are not in italics.  ­­This change should come from an error during the change from .docx to .pdf format  of the manuscript. In the final file of the publishing manuscript, you can be sure that we are going to verify that all the scientific names of plant species are in italic.

Regarding observation number 2:

  1. When describing fleshy fruit cuticle compositions, different cultivars of certain fruit, like apple, present obvious differences in cuticle components. What are the similarities of cultivars sharing similar components? Do they share similar geographical location?

It is a very interesting observation. Experimental evidences suggests that cuticle composition patterns are influenced by the geographical location and environmental conditions where fruit cultivars evolved [1]. In fact, it has been proposed that cuticle composition patterns could be used to determine the geographical origin of fruit cultivars [1]. Nevertheless, for most of the cultivars analyzed in the studies included in this review, information about the geographical location is not provided, or it is not discussed. For instance, ‘Royal Gala’, ‘Granny Smith’, ‘Pink Lady’, and ‘Red Delicious’ apple cultivars, which have a very similar cuticle composition, were purchased at a local market in the city of Santa María, Brazil, but information about the origin of that cultivars was not provided [2].

Most of the studies included in this review that reported similarities between cultivars were carried out from cultivars belonging to the same experimental field with the same environmental conditions, or experimental fields that are in the same country. The origin of apple fruit cultivars ‘Florina’ and ‘Prima’ is France and USA, respectively [3]. Nevertheless, for the analysis cited in this review, both were grown and harvested in an orchard at the Experimental Research Farm of Szent István University, Soroksár [3]. Therefore, the similarities  between different cultivars of certain fruit could be due to the fact that they are in the same environmental conditions in the experimental field. Alternatively, these similarities could be due to the close genetic relationship between cultivars.

‘Golden Delicious’ and ‘Red Star’ apple cultivars were harvested at Shunyi orchard, whereas ‘Stark’, ‘Golden’ and ‘Mutsu’ were harvested at Changping orchards, in Beijing, China. Nevertheless, similarities between cultivars regarding their geographical location is not discussed [4]. However, more studies focused on this issue should be carry out since it is a very interesting observation.

Regarding observation number 3:

  1. Are the cuticle compositions related with the original flower organ where the fleshy fruit develop from? If it is related, it would be better to describe the original flower organs of each fleshy fruit.

It is a very interesting question. The cuticle composition varies between different plant organs of the same species such as stems, leaves,  fruits, and between different flower organs [5, 6]. This specific composition is related to the physical characteristics of the organ and its physiological function [6]. It makes sense to think that the cuticle composition of fruits is related to the tissue from which they originated, nevertheless, there are not studies available trying to establish a relation between the cuticle composition of fleshy fruits and the chemical composition of the original flower organ where they develop from. On the other hand, cuticle composition analysis of fleshy fruits carried out during a very early stage of development gives us an approach to the cuticle composition of the original floral organ.

Regarding to observation number 4:

  1. When describing the molecular pathway involved, please make it more clear in which species the molecular knowledge was obtained.

Thank you for your valuable observation. Almost all of these discoveries about the genes playing a role in cuticle biosynthesis have been carried out in the model plants Arabidopsis and tomato. Excellent reviews about the molecular pathway of cuticle biosynthesis in plants are published and we had included the references  in this manuscript. By other side, we decided to create a brief overview of the core molecular pathway, with an emphasis on the molecular knowledge that has been generated with experiments in fruits. Also, the different  species are mentioned in the text. Nevertheless, to improve this section and attend to your valuable suggestion, when the molecular knowledge was not obtained from Arabidopsis, we have included in the  species out of which the molecular knowledge was obtained.

References

  1. de Rijke, E., Fellner, C., Westerveld, J. Lopatka, C. Cerli, K. Kalbitz & C. G. de Koster. Determination of n-alkanes in C. annuum(bell pepper) fruit and seed using GC-MS: comparison of extraction methods and application to samples of different geographical origin. Anal Bioanal Chem 2015, 407, 5729–5738. https://doi.org/10.1007/s00216-015-8755-6

  1. Klein, B., Thewes, F. R., Anese, R. d. O., Brackmann, A., Barin, J. S., Cichoski, A. J. and Wagner, R. Development of dispersive solvent extraction method to determine the chemical composition of apple peel wax. Food Research International 2019, 116,, 611-619, doi:https://doi.org/10.1016/j.foodres.2018.08.080.

  1. Leide, J., Xavier de Souza, A., Papp, I. and Riederer, M. Specific characteristics of the apple fruit cuticle: Investigation of early and late season cultivars ‘Prima’ and ‘Florina’ (Malus domestica Borkh.). Scientia Horticulturae 2018, 229, 137-147, doi:10.1016/j.scienta.2017.10.042.

  1. Chai, Y., Li, A., Chit Wai, S., Song, C., Zhao, Y., Duan, Y., Zhang, B. and Lin, Q. Cuticular wax composition changes of 10 apple cultivars during postharvest storage. Food Chem 2020, 324, 126903, doi:10.1016/j.foodchem.2020.126903.

  1. Diarte, C., Xavier de Souza, A., Staiger, S., Deininger, A.-C., Bueno, A., Burghardt, M., Graell, J., Riederer, M., Lara, I. and Leide, J. Compositional, structural and functional cuticle analysis of Prunus laurocerasus L. sheds light on cuticular barrier plasticity. Plant Physiology and Biochemistry 2021, 158, 434-445, doi:https://doi.org/10.1016/j.plaphy.2020.11.028.

  1. Huang, H., Hu, Y., Wang, L., Li, F., Shan, Y., Lian, Q., and Jiang, Y. Comparative profiles of the cuticular chemicals and transpiration barrier properties in various organs of Chinese flowering cabbage and Chinese kale. Physiologia Plantarum 2022, 174, e13650 , https://doi.org/10.1111/ppl.13650. Volume174, Issue2.